# D2SA: Dual-Stage Distribution and Slice Adaptation for Efficient Test-Time Adaptation in MRI Reconstruction

**Lipei Zhang**[1]*, **Rui Sun**[2,3]*, **Zhongying Deng**[1], **Yanqi Cheng**[1],
**Carola-Bibiane Schönlieb**[1], **Angelica I Aviles-Rivero**[4]

[1] Department of Applied Mathematics and Theoretical Physics, University of Cambridge
[2] Shenzhen Future Network of Intelligence Institute
and Guangdong Provincial Key Laboratory of Future Networks of Intelligence,
The Chinese University of Hong Kong (Shenzhen)
[3] School of Science and Engineering, The Chinese University of Hong Kong (Shenzhen)
[4] Yau Mathematical Sciences Center, Tsinghua University

## Abstract

Variations in Magnetic resonance imaging (MRI) scanners and acquisition protocols cause distribution shifts that degrade reconstruction performance on unseen data. Test-time adaptation (TTA) offers a promising solution to address this discrepancies. However, previous single-shot TTA approaches are inefficient due to repeated training and suboptimal distributional models. Self-supervised learning methods may risk over-smoothing in scarce data scenarios. To address these challenges, we propose a novel Dual-Stage Distribution and Slice Adaptation (D2SA) via MRI implicit neural representation (MR-INR) to improve MRI reconstruction performance and efficiency, which features two stages. In the first stage, an MR-INR branch performs patient-wise distribution adaptation by learning shared representations across slices and modelling patient-specific shifts with mean and variance adjustments. In the second stage, single-slice adaptation refines the output from frozen convolutional layers with a learnable anisotropic diffusion module, preventing over-smoothing and reducing computation. Experiments across five MRI distribution shifts demonstrate that our method can integrate well with various self-supervised learning (SSL) framework, improving performance and accelerating convergence under diverse conditions.

## 1 Introduction

Magnetic resonance imaging (MRI) captures detailed tissue structures using k-space sampling. In clinical practice, MRI is often under-sampled to accelerate scan time and reduce patient burden. However, under-sampling results in an ill-posed inverse problem, making accurate MRI reconstruction challenging [23]. Traditional compressed sensing techniques attempt to address this through iterative reconstruction algorithms [4, 5, 3, 28], but these methods are computationally expensive and less accurate. Recent advances in deep learning have significantly improved both reconstruction speed and quality by learning direct mappings from raw data [18], such as unrolled networks [27], plug-and-play frameworks [1], and diffusion models [8].

Despite these advancements, deep learning models struggle with adapting to diverse clinical scenarios due to two primary challenges. Firstly, limited MRI data for model adaptation: MRI datasets are difficult to collect, making it challenging to generalise deep models without overfitting. Secondly,

---

*These authors contributed equally.

39th Conference on Neural Information Processing Systems (NeurIPS 2025).

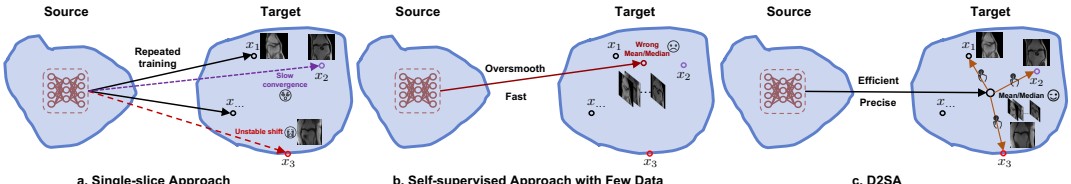

Figure 1: Illustration of TTA strategies for MRI reconstruction under distribution shifts. (a) Single-slice methods require repeated training and are often unstable.(b) Self-supervised approaches with limited data may oversmooth or converge to incorrect mean/median of target domain. (c) **D2SA** first performs efficient, patient-level adaptation to unknown target distributions (blue path), followed by optional slice-level refinement guided by requirement from clinician (orange path and 👆).

distribution shifts between training and test data: In real-world deployment, MRI scans may be acquired under different conditions (e.g., scanner types, patient demographics), causing performance degradation due to mismatched data distributions between training and test sets [10, 11]. An ideal MRI reconstruction model should therefore balance three key goals for overcoming distribution shifts: 1) *Strong adaptation to new distributions* – maintaining high performance despite distribution shifts. 2) *Robustness to limited data* – preventing overfitting in data-scarce scenarios. 3) *Fast convergence* – minimising adaptation time at test time.

Most existing methods focus primarily on distribution generalisation but fail to optimise all three goals simultaneously. Test-time adaptation (TTA) techniques partially address this, i.e., they mitigate the distribution shift by updating models on the fly using only test data. Besides handling distribution shifts, batch-based TTA methods (e.g., Noiser2noise [12], FINE [38], SSDU [36]) further enforce self-supervised learning across multiple slices to facilitate fast convergence. However, this batch-wise approach may overfit shared features across slices while ignoring slice-specific variations, leading to over-smoothed reconstructions. Conversely, single-slice-based TTA methods [41, 34, 35] improve fine-grained adaptation but require repeated optimisation cycles, significantly increasing computational overhead. More recent diffusion-based models [2, 9] generate realistic slices for adaptation but are computationally expensive and prone to overfitting on smaller datasets.

To effectively balance all three goals, we propose Dual-Stage Distribution and Slice Adaptation (D2SA). D2SA leverages both patient-wise and slice-wise adaptation through a two-stage process. The first stage models single patient distribution using a small number of slices as prior knowledge. The second stage utilises this learned prior for fast adaptation to each slice, and further introduces an anisotropic diffusion (AD) module to enhance denoising [21, 7] while preventing over-smoothing the structural details. It thus achieves fast adaptation with high reconstruction quality. Both stages treat each MRI slice as a continuous function rather than a static matrix, drawing inspiration from Functa [13] and implicit neural representations (INRs)[25]. This function-based perspective allows us to interpret distribution shifts as small function-level variations, e.g., functions with different mean/variance variables in the feature space. Owing to the adaptive mean/variance, this function-centric approach can be efficiently adapted to new distributions without the need for extensive data for retraining. It also enables the plug-in of networks at test time, thus highly flexible. Our novel approach ensures fast convergence, robustness to limited data, and strong generalisation to new distributions, addressing a critical gap in MRI reconstruction research. Our contributions are:

- **Functional-Level Patient Adaptation.** We develop an INR-based strategy that learns a patient's distribution from a small number of slices, with the INRs trained to capture individualised mean and variance shifts for the second-stage fast adaptation.

- **Structural-Preserving Single-Slice Refinement.** After modelling patient-level shifts, the pre-trained INR network rapidly refines each slice. We introduce a learnable Anisotropic Diffusion (AD) module to maintain structural fidelity, reduce over-smoothing, and limit computation by freezing the main convolutional layers.

- **Extensive Validation.** We evaluate D2SA on five distribution shift scenarios, using both UNet [26] and a variational network [31]. Results demonstrate robust and efficient reconstruction across diverse clinical conditions.

## 2 Related Work

**Test-Time Adaptation (TTA) in Medical Imaging.** TTA tackles distribution shifts by adapting pre-trained models using unlabelled test data [22]. A key challenge is constructing supervision signals without ground truth, typically addressed via consistency regularisation or self-supervised losses. Consistency-based methods enforce stable predictions under perturbations. For instance, PINER [30] leverages implicit neural representations (INRs) to select resolution-consistent CT slices, while steerable diffusion models [2] ensure realistic reconstructions. Self-supervised approaches define pretext tasks such as contrastive learning [19] or rotation prediction [17]. DIP-TTT [11] applies self-supervision for slice-wise reconstruction under shifts, and Meta-TTT [34] incorporates meta-learning to improve generalisation. In contrast to computationally expensive TTA methods, we propose a dual-stage TTA framework that first performs patient-level adaptation to improve the efficiency and robustness of per-slice refinement.

**Implicit Neural Representations (INRs).** INRs encode data as continuous functions, enabling compact and flexible learning. Functa [14] embeds entire datasets as INRs for function-level learning, while DeepSDF [25] uses latent-conditioned autodecoders to model 3D shape fields. Biomedical INRs have been used to represent detailed structures like airway trees [40], allowing for effective batch optimisation. Among various designs [15], SIREN [29] remains a strong choice for high-frequency data due to its sine activation, and forms the basis of our patient-wise adaptation module.

## 3 Problem Setup

First, MRI reconstruction is an inverse problem where the goal is to recover $x^* \in \mathbb{C}^N$ from undersampled measurements $y \in \mathbb{C}^M$ with $M \ll N$: $y = A\,x^* + \epsilon$, where $A$ is the measurement operator, and $\epsilon$ represents noise. In multi-coil MRI, the acquired measurements for each coil $i$ follow:

$$y_i = M\,F\,S_i\,x^* + \epsilon, \quad i = 1, \ldots, n_c, \tag{1}$$

where $S_i$ denotes the coil sensitivity map, $F$ is the 2D Fourier transform. $M$ is the undersampling mask which can be the 1D cartesian mask, or others [37]. The individual coil images $x_i = F^{-1}y_i$ are then combined via root-sum-of-squares to reconstruct $x$.

Reconstruction is framed as an optimisation problem:

$$\hat{x} = \arg\min_x \ \frac{1}{2}\|A\,x - y\|_2^2 + \lambda R(x), \tag{2}$$

where $R(x)$ encodes prior knowledge (e.g., wavelet $\ell_1$, total variation, or CNN-based priors), and $\lambda$ controls the balance between data fidelity and regularisation. However, standard reconstruction models assume a fixed distribution during testing, limiting their ability to generalise to new datasets or acquisition conditions.

Domain shifts from scanners, anatomy, or acquisition protocols degrade performance. Existing TTA methods can address this but they rely on repeated single-slice training [11] or self-supervised learning on large datasets [12, 38, 36], lacking stability in data-scarce scenarios. To address this, we introduce a D2SA that first learns patient-wise distributions explicitly for better initialisation, enabling more stable and efficient refinement in the second stage.

## 4 Proposed Method

To address domain shifts efficiently, we propose D2SA, a dual-stage test-time adaptation (TTA) framework that avoids large datasets and slow repeated single-slice training. As illustrated in Figure 2, D2SA consists of: (1) **Patient-wise Distribution Adaptation**, where MR-INR captures shared representations across slices and estimates variance ($\alpha$) and mean ($\beta$) shifts; and (2) **Single-Slice Refinement (SST)**, which refines each slice using a learnable anisotropic diffusion (AD) module with frozen convolutional layers. This design enables efficient adaptation with improved initialization.

### 4.1 Functional-Level Patient Adaptation

In Figure 2, the MR-INR branch models the structure and distribution of patient-wise data. Inspired by Functa [13] and DeepSDF [25], which use INRs to encode data as continuous functions, our

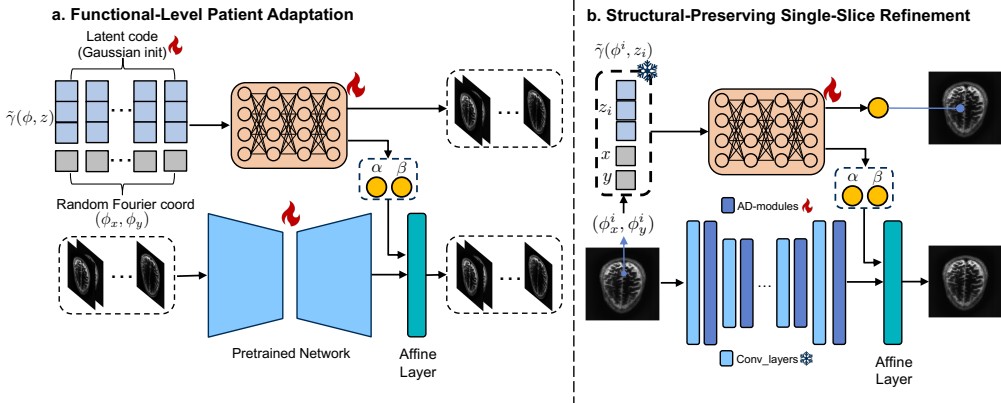

Figure 2: Overview of the proposed two-stage D2SA framework. (a) Functional-Level Patient Adaptation: An INR with a Gaussian-initialized latent code and random Fourier coordinates captures patient-level mean/variance shifts. The 🔥 indicates trainable modules, including the "pretrained" network and the affine layer. (b) Structural-Preserving Single-Slice Refinement (SST): The main convolutional layers and learned latent code are frozen ❄️., while a learnable Anisotropic Diffusion (AD) module and the INR refine individual slices, preserving structural details and finalising outputs via the affine layer.

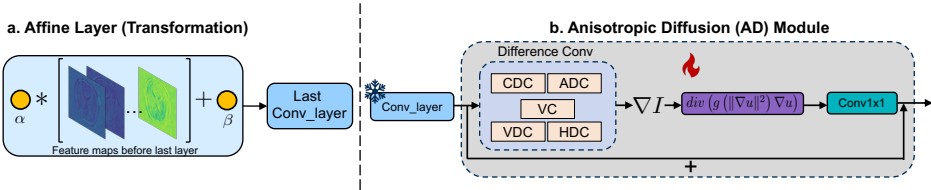

Figure 3: (a) Learnable affine transform scales feature maps by $\alpha$ and $\beta$ before the final layer. (b) Learnable AD module 🔥 refines images while preserving structures, with frozen convolution ❄️.

approach shifts from learning on discrete datasets to learning in function spaces. This enables efficient adaptation to new unknown distributions, better handling of few-shot scenarios, and improved patient-wise learning capabilities.

In Figure 2.a, each slice in this patient set can we assume the prior distribution over a 1D latent code $z_i$ as zero-mean multivariate-Gaussian with a spherical covariance $\sigma^2 I$. In this work, $\sigma$ is set to 0.01. The random Fourier coordinates will be calculated by geometric coordinates $\phi$ of each slice. The input $\tilde{\gamma}$ for MR-INR is from concatenation of $z_i$ and $\phi$.

$$\tilde{\gamma}(\phi, z_i) = [z_i, \cos(2\pi B\phi), \sin(2\pi B\phi)], \tag{3}$$

where the transformation matrix $B$ is sampled from a Gaussian distribution $\mathcal{N}(0, \omega^2)$.

After this step, we use the standard batch training protocol for all slices in each patient. In the MR-INR branch, the corresponding latent code and Fourier coordinates are modulated and passed through a SIREN [29] network $f_\theta$ architecture. The ability of SIREN and Fourier feature [32] to efficiently model target representation and stability has been shown in [15]. This MR-INR branch can be formed as:

$$\begin{aligned}
[\hat{x}, \alpha, \beta] &= f_\theta(\tilde{\gamma}(\phi^i, z)) = W_n(\Gamma_{n-1} \circ \Gamma_{n-2} \circ \cdots \circ \Gamma_0)(\tilde{\gamma}(\phi^i, z)) + b_n, \\
h^{(i+1)} &= \Gamma_i(h^{(i)}) = \sin(W_i h^{(i)} + b_i),
\end{aligned} \tag{4}$$

Here, $\Gamma_i : \mathbb{R}^{M_i} \to \mathbb{R}^{N_i}$ represents the $i^{th}$ transformation layer. Each layer applies an affine transformation with weight matrix $W_i \in \mathbb{R}^{N_i \times M_i}$ and bias $b_i \in \mathbb{R}^{N_i}$, followed by a sine activation function. The final layer produces $[\hat{x}, \alpha, \beta]$ through three output heads. $\hat{x}$ with dimension $(B, 2, H, W)$, represents the predicted pixel intensity for MRI reconstruction. $\alpha$ variance shifts and $\beta$ nonzero-mean shifts, with dimension $(B, C, H, W)$, modulate feature maps before the final layer via an affine transformation, as shown in Figure 3.a, where $C$ is the number of feature channels. This formulation

enables a shared base network to model common structures while adapting to patient-specific variations, ensuring a compact and efficient solution for TTA. Meanwhile, in the first stage, under-sampled MR images from the target domain are input into the network $g_\delta$, initialised with source domain pre-trained weights, for TTA ($g_\delta \to g_{\delta+\Delta}$). Here, feature maps before last layer are extracted and adjusted via the affine transformation using $\alpha$ and $\beta$, as illustrated in Figure 3.a.

The predicted MR image from this branch is used to compute the self-supervised loss $\mathcal{L}_{\text{self}}$ such as Noiser2noise [12, 24], SSDU [36] and fidelity-based FINE [38]. The $\mathcal{L}_{\text{self}}$ combines with other two loss from MR-INR for joint optimisation. For MR-INR, we adopt a joint optimisation strategy similar to the auto-decoder framework [25], optimising both latent codes and network parameters. The optimisation for the first stage is formulated as:

$$
\begin{aligned}
\hat{\theta}, \hat{z}, \hat{\Delta} = \arg \min_{\theta, z, \Delta} \lambda_{\text{INR}} \sum_{(x_j, z_j, y_j) \in X} \mathcal{L}_1(Af(\tilde{\gamma}(\phi^j, z), \theta), y_j) \\
+ \lambda_{\text{reg}} \frac{1}{\sigma^2} \|z\|_2^2 \quad + \lambda_{\text{self}} \sum_{(x_j) \in X} \mathcal{L}_{\text{self}}(g(x_j, \alpha, \beta, \delta + \Delta)).
\end{aligned}
\tag{5}
$$

Here, $\mathcal{L}_1(Af(\tilde{\gamma}(\phi^j, z), \theta), y_j)$ ensures INR predictions aligning with MR signal consistency. The regularisation term $\frac{1}{\sigma^2}\|z\|_2^2$ constrains sparsity of the latent codes and prevents overfitting. The final term updates weights $\delta + \Delta$ to adapt to patient-specific variations using self-supervised loss. $\lambda_{\text{self}}$, $\lambda_{\text{INR}}$ and $\lambda_{\text{reg}}$ are used for balancing every loss contribution.

## 4.2 Structural-Preserving Single-Slice Refinement Training

To refine MRI reconstruction at the slice level, we design a new SST strategy for fine-grained adjustments. Unlike patient-wise adaptation, which learns shared representations across slices, this stage optimises each slice independently to capture localised variations, as shown in Figure 2.b. The latent codes are frozen to preserve the learned global prior information from patient-wise training. This prevents instability and avoids overfitting to slice-specific noise. Instead, the SIREN weights remain trainable, allowing the model to refine its implicit function for each slice. The affine modulation parameters $\alpha, \beta$ continue adjusting the final feature maps via scaling and shifting.

In the other branch, we adopt a similar DIP-based TTT strategy [11] for SST. This approach leverages CNNs' strong image priors for structural preservation and optimises a self-supervised loss $\mathcal{L}_{\text{self}}$ on under-sampled test measurements. To improve efficiency, we freeze all convolutional layers except the final one and transpose convolutions, reducing unnecessary updates and accelerating optimisation.

A key challenge in batch training is its tendency to learn mean or median representations, leading to over-smoothing that can obscure fine textures and edges. This is critical in MRI, where structural details must be preserved. To address this, we introduce an Anisotropic Diffusion (AD) module, inspired by its shape-preserving properties in image denoising [7, 6]. As shown in Figure 3.b, the AD module refines structural details while suppressing noise by integrating diffusion filtering into the adaptation process. Given an set of feature $u$, the AD equation is:

$$
\left(\frac{\partial u}{\partial t}\right) = \text{div}\left(g(|\nabla u|)\nabla u\right); \quad g(|\nabla u|) = \frac{1}{1 + \frac{|\nabla u|^2}{k^2}}
\tag{6}
$$

When the gradient magnitude is small ($|\nabla u| \to 0$), the diffusion coefficient $g$ approaches 1, leading to isotropic smoothing similar to Gaussian filtering. Near object boundaries, where $|\nabla u| \to 1$, $g$ approaches 0, preserving fine details. This allows AD to suppress noise effectively while keeping sharp edges, making it well-suited for edge-aware regularisation in reconstruction.

We enhance traditional convolutions by integrating difference-based operators [7] that explicitly encode gradient information $\nabla u$. Five types of convolutions are introduced: Vanilla Convolution (VC), Central Difference Convolution (CDC), Angular Difference Convolution (ADC), Horizontal Difference Convolution (HDC), and Vertical Difference Convolution (VDC). These capture multiple directional gradients, incorporating concepts from Sobel, Prewitt, and Scharr filters directly into the convolution process [7]. The convolution operation is formulated as:

$$
\nabla u = F_{\text{out}} = \text{DConv}(F_{\text{in}}) = \sum_{i=1}^{5} F_{\text{in}} * K_i = F_{\text{in}} * K_{\text{cvt}},
\tag{7}
$$

where $F_{\text{in}}$ and $F_{\text{out}}$ represent input and output feature maps, respectively. Instead of separate convolutions, we merge all five kernels $K_i$ into a single equivalent kernel $K_{\text{cvt}}$ using a re-parameterisation technique. To improve efficiency, we reduce the number of output feature maps to 1/4 of the original channels, ensuring compact gradient extraction while minimising redundancy.

In calculation of AD equation (6), the computed $\nabla u$ is used to determine the diffusion coefficient $g$, while the divergence $\text{div}(\cdot)$ is approximated via a 2D Laplacian kernel, which is more efficient to preserve spatial information than standard finite difference methods [39]. The output of the first equation in (6) is restored to its original feature map dimensions using a $1 \times 1$ convolution. Setting the diffusion step size $\Delta t = 1$ in (6), the updated feature maps are:

$$u_{i+1} = u_i + \Delta t \cdot Conv_{1\times 1}(\text{div}\,(g(|\nabla u_i|)\nabla u_i)). \tag{8}$$

In this stage, we optimise the weights in MR-INR and the original network with AD module. The final loss function for the second step is:

$$\hat{\theta}, \hat{\Delta} = \arg\min_{\theta,\Delta} \sum_{(x_j,y_j)\in X} \lambda_{\text{INR}} \underbrace{\mathcal{L}_1\big(Af(\tilde{\gamma}(\phi^j,\hat{z}),\theta), y_j\big)}_{\text{MR-INR consistency loss}} + \sum_{(y_j)\in X} \lambda_{\text{self}} \underbrace{\frac{|y_j - Ag(\mathbf{A}^\dagger y_i, \delta + \Delta)|_1}{|y_j|_1}}_{\text{Self-sup loss}}. \tag{9}$$

The first term is for measurement consistency, ensures that the MR-INR branch reconstructs MRI images accurately. The second term, Self-Supervised loss, refines the prediction using measurement consistency. This formulation enables adaptive refinement while preserving prior knowledge learned from the first stage. $\lambda_{\text{self}}$ and $\lambda_{\text{INR}}$ are used for balancing every loss contribution.

### 4.3 Mathematical Analysis of Affine Adaptation

To motivate our use of affine transformations during test-time adaptation, we analyze a simplified setting under distribution shift. While our method uses nonlinear system, this linear case offers insights into how the learned parameters $\alpha$ and $\beta$ operate under such shift. Consider the test distribution:

$$Q: \quad \mathbf{y} = \mathbf{x} + \mathbf{z}, \quad \mathbf{x} = \mathbf{U}\mathbf{c} + \mu_Q, \quad \mathbf{c} \sim \mathcal{N}(0, I), \quad \mathbf{z} \sim \mathcal{N}(0, s^2\mathbf{I}). \tag{10}$$

Here, $\mathbf{U} \in \mathbb{R}^{n\times d}$ is an orthonormal basis of the signal subspace, and $\mu_Q$ encodes the mean shift. Our goal is to estimate $\mathbf{x}$ under this shift. The optimal TTA estimator and self-supervised loss are next:

**Proposition 1.** *An affine estimator of the form* $\hat{\mathbf{x}} = \alpha\mathbf{U}\mathbf{U}^T\mathbf{y} + \beta$ *minimizes the following self-supervised loss:*

$$L_{SS}(\alpha, \beta) = \mathbb{E}_Q\left[\left\|\mathbf{y} - \alpha\mathbf{U}\mathbf{U}^T\mathbf{y} - \beta\right\|_2^2\right] + \frac{2\alpha d}{n-d}\mathbb{E}_Q\left[\left\|(\mathbf{I} - \mathbf{U}\mathbf{U}^T)\mathbf{y}\right\|_2^2\right]. \tag{11}$$

**Theorem 1.** *Minimizing* $L_{SS}(\alpha, \beta)$ *yields optimal parameters by solving the first-order conditions. The gradients are:*

$$\frac{\partial L_{SS}}{\partial \beta} = 2(\beta - \mu_Q), \quad \frac{\partial L_{SS}}{\partial \alpha} = -2d(1-\alpha) + 2\alpha d s^2.$$

*Solving these gives the optimal solutions* $\alpha^* = \frac{1}{1+s^2}$ *and* $\beta^* = \mu_Q$.

These parameters decouple the effects of noise and mean shift: $\alpha^*$ corrects variance, and $\beta^*$ aligns the mean. They are learned by minimizing the self-supervised loss under the test distribution. This analysis provides a clear intuition for our design: although the full model is nonlinear, we apply the linear affine adaptation in feature space of MR-INR. More detailed proof and derivations are in the Appendix. Next, our empirical results further confirm the robustness of this effective TTA approach.

## 5 Experimental Settings

### 5.1 Datasets and Experimental Settings

We evaluate on multi-coil MRI data from **fastMRI** [37] (knee, brain) and **Stanford** [16]. Each experiment defines a source distribution $\mathcal{S}$ and target distribution $\mathcal{T}$, measuring performance via

SSIM, PSNR, and LPIPS. We consider two baselines: (1) **U-Net** [26]: 8 layers, 64 channels, trained with Adam [20] at learning rate $10^{-5}$; (2) **VarNet** [31]: 12 cascades, 18 channels, trained with Adam at $10^{-4}$. All other training settings follow [11], using combination of supervised and self-supervised losses. We simulate $4\times$ undersampling with 1D random Cartesian masks and 8% auto-calibration lines, estimating sensitivity maps via ESPiRiT [33]. We examine five domain shifts: *anatomy*, *dataset*, *modality*, *acceleration*, and *sampling*, evaluating both out-of-distribution ($\mathcal{S} \rightarrow \mathcal{T}$) and in-distribution performance (see Appendix).

**Anatomy Shift.** Following in [11], U-Net and VarNet are trained on fastMRI knee data as the source domain ($\mathcal{S}$) and evaluated on fastMRI AXT2 brain data as the target domain ($\mathcal{T}$). For TTA evaluation, we randomly select 10 subjects, resulting in 110 AXT2 brain slices subsampled at $4\times$.

**Dataset Shift.** Following [11], we train both models on Stanford knee data ($\mathcal{S}$) and evaluate on fastMRI knee data ($\mathcal{T}$). We sample 20 patients from fastMRI, yielding 400 knee slices under the same $4\times$ subsampling ratio for TTA evaluation.

**Modality Shift.** U-Net and VarNet are trained on fastMRI AXT2 brain slices ($\mathcal{S}$) and tested on AXT1PRE slices ($\mathcal{T}$), following the setup in [11]. We randomly select 10 patients, yielding 110 AXT1PRE brain slices with $4\times$ subsampling for TTA.

**Acceleration Shift.** Models are trained on fastMRI knee measurements acquired with $2\times$ acceleration ($\mathcal{S}$) and tested on the same set of knee slices with $4\times$ acceleration ($\mathcal{T}$), as in [11]. We evaluate TTA on 400 slices sampled from 20 patients.

**Sampling Shift.** The uniform sampling presents more coherent artifacts that are more challenging to handle. We also train on fastMRI AXT2 brain data subsampled using a random 1D Cartesian mask at $4\times$ acceleration ($\mathcal{S}$)[11], and evaluate on the same AXT2 slices sampled with a uniform 1D mask at the same acceleration rate ($\mathcal{T}$). For TTA, we randomly select 10 patients, totaling 110 slices.

Additional details of stage-1 and stage-2 training procedures are in the Appendix. **Appendix** is provided as a **separate** file in supplementary materials.

## 5.2 Compared Methods

We compare D2SA with four representative test-time adaptation (TTA) baselines. **DIP-TTT** [11] performs single-slice adaptation using Deep Image Prior (DIP). **FINE** [38] is a batch-level TTA approach based on fidelity constraints. **Noiser2noise (NR2N)** [12, 24] and **SSDU** [36] are self-supervised methods that operate in a patient-wise manner.

DIP-TTT follows its original setting, while FINE, NR2N, and SSDU are trained using the same configuration as the first stage of our method. To assess the benefit of MR-INR, we integrate it into FINE, NR2N, and SSDU for patient-wise adaptation. All resulting pretrained models—with and without MR-INR—are then used in the second-stage single-slice refinement under the same setup as our stage 2. Models without MR-INR adopt only self-supervised loss, similar to DIP-TTT.

We also include **ZS-SSL** [35], an augmentation-based self-supervised single-slice TTA method. Our preliminary results show that it is compatible only with unrolled networks such as VarNet, and fails to generalise to end-to-end U-Net architectures. Detailed comparisons are provided in the Appendix. All experiments were run on a single NVIDIA RTX 3090 GPU. For timing, Stage 1 inference time is measured as the total duration of 25 fixed training epochs. Stage 2 (SST) time is computed as the sum of per-slice training durations until early stopping, using the same validation-based strategy as in [11]. The final reported time combines both stages.

## 6 Results & Discussion

**Main results.** Tables 1 and Figure 4 summarize the average SSIM, PSNR, LPIPS, and adaptation time for U-Net and VarNet under five domain shifts: *Anatomy*, *Dataset*, *Modality*, *Acceleration*, and *Sampling*. Across most settings in Table 1, +MR-INR+SST achieves consistent improvements over baselines. For example, in the acceleration shift, FINE+MR-INR (VarNet) improves SSIM from 0.696 to 0.791 and PSNR from 21.39 to 25.30. While MR-INR introduces a modest runtime overhead. +MRI-INR+SST performances rival or exceed DIP-TTT in multiple cases (e.g., SSDU on anatomy shift, NR2N on dataset shift), while significantly reducing adaptation time (e.g., 17.1 vs. 52.5 mins/patient in anatomy shift). These results demonstrate the strong synergy between patient-wise MR-INR adaptation and single-slice SST refinement.

| Method (VarNet) | Anatomy Shift (S: Knee, T: Brain) | Dataset Shift (S: Stanford, T: fastMRI) | Modality Shift (S: AXT2, T: AXT1PRE) | Acceleration Shift (S: 2x, T: 4x) | Sampling Shift (S: Random, T: Uniform) |
|---|---|---|---|---|---|
| Zero-filling | 0.737/24.50/0.327/- | 0.747/24.33/0.359/- | 0.747/25.71/0.350/- | 0.754/23.37/0.396/- | 0.766/26.28/0.338/- |
| Non-TTA | 0.799/23.16/0.371/- | 0.706/22.35/0.365/- | 0.796/23.54/0.379/- | 0.761/23.04/0.372/- | 0.111/16.00/0.594/- |
| DIP-TTT | 0.878/27.67/0.312/52.5 | 0.798/28.02/0.292/41.8 | 0.867/28.33/0.337/71.6 | 0.815/28.25/0.285/137.2 | 0.771/27.65/0.254/38.9 |
| FINE | 0.820/24.01/0.343/3.9 | 0.789/26.26/0.311/6.6 | 0.821/26.18/0.369/3.5 | 0.696/21.39/0.342/6.2 | 0.669/21.18/0.365/3.7 |
| FINE+MR-INR | 0.862/26.45/0.328/4.7 | 0.795/26.44/0.306/6.9 | 0.830/26.58/0.369/4.4 | 0.791/25.30/0.310/7.5 | 0.789/25.10/0.339/4.1 |
| FINE+SST | 0.862/27.57/0.311/53.5 | 0.794/27.72/0.294/20.3 | 0.857/28.08/0.345/79.8 | 0.823/28.17/0.288/63.8 | 0.748/25.18/0.276/48.3 |
| FINE+MR-INR+SST | 0.882/27.68/0.311/17.1 | 0.808/28.72/0.286/18.2 | 0.867/28.32/0.337/21.8 | 0.829/28.64/0.276/44.5 | 0.824/28.49/0.232/22.1 |
| NR2N | 0.827/23.95/0.334/4.9 | 0.798/26.59/0.299/6.8 | 0.827/25.60/0.368/4.1 | 0.718/20.97/0.327/6.6 | 0.661/21.15/0.379/4.3 |
| NR2N+MR-INR | 0.868/26.41/0.321/5.1 | 0.806/26.95/0.294/7.1 | 0.833/26.44/0.369/4.7 | 0.806/25.42/0.291/7.6 | 0.786/25.24/0.366/4.5 |
| NR2N+SST | 0.883/27.72/0.307/63.9 | 0.798/27.78/0.293/25.3 | 0.860/28.17/0.341/80.2 | 0.822/28.09/0.291/69.2 | 0.771/26.46/0.276/57.1 |
| NR2N+MR-INR+SST | 0.884/27.81/0.306/18.7 | 0.812/28.76/0.281/20.4 | 0.869/28.36/0.336/20.3 | 0.826/28.89/0.273/43.1 | 0.822/28.40/0.241/25.2 |
| SSDU | 0.738/20.87/0.375/5.1 | 0.737/20.43/0.349/7.2 | 0.746/22.59/0.391/4.4 | 0.556/16.93/0.421/7.4 | 0.483/17.53/0.415/4.0 |
| SSDU+MR-INR | 0.821/23.25/0.350/5.3 | 0.764/21.67/0.339/7.5 | 0.796/24.09/0.390/4.8 | 0.728/19.57/0.358/7.9 | 0.554/18.88/0.409/5.3 |
| SSDU+SST | 0.879/27.65/0.310/68.3 | 0.789/26.79/0.299/24.9 | 0.857/28.07/0.343/93.4 | 0.803/28.06/0.293/134.2 | 0.736/24.84/0.288/50.5 |
| SSDU+MR-INR+SST | 0.882/27.66/0.307/18.5 | 0.808/28.16/0.290/21.4 | 0.863/28.09/0.342/29.3 | 0.826/28.62/0.286/45.2 | 0.800/28.21/0.254/25.8 |

Table 1: Performance comparison of VarNet methods under different domain shifts. Each cell presents ((**SSIM ↑ / PSNR ↑ / LPIPS ↓ / Time (mins/patient) ↓**). The family of proposed methods incorporates a self-supervised learning framework, combining MR-INR-based patient-wise adaptation with single-slice refinement using pre-trained patient-wise models.

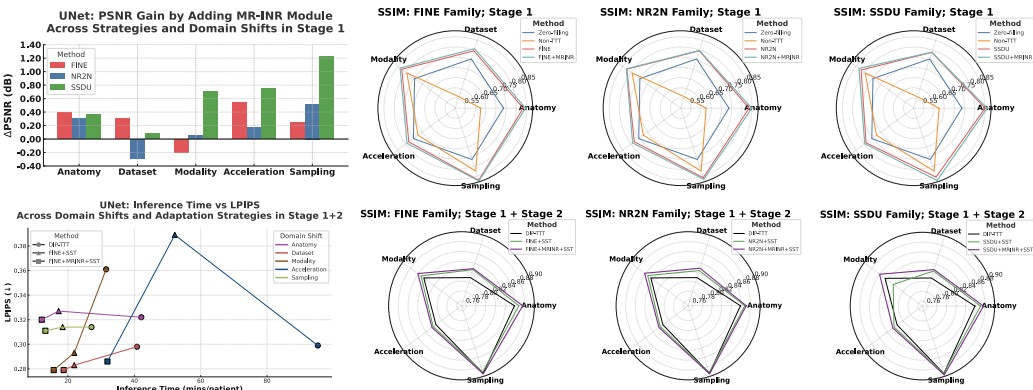

Figure 4: Performance Analysis of U-Net under Different Domain Shifts. Top-left: PSNR gain from adding MR-INR across FINE, NR2N, and SSDU in stage 1, with the largest gain in modality shift in most strategies. Bottom-left: LPIPS vs. inference time trade-off showing that +MR-INR+SST achieves higher quality with reduced time comsumption compared to DIP-TTT and normal +SST. Right: SSIM across five domain shifts for each SSL family, visualized patient-level (Stage 1) and after slice refinement (Stage 1 + Stage 2). MR-INR consistently improves performance across domains, and combining it with SST further enhances SSIM.

Figure 4 further illustrates the effectiveness of MR-INR. The top-left plot shows PSNR consistently increases across most TTA families, with the largest gain in the sampling shift. The bottom-left trade-off curve shows MR-INR+SST achieves lower LPIPS with lower cost than DIP-TTT. Radar plots confirm SSIM gains from Stage 1 (MR-INR) and further improvements when combined with Stage 2 (SST). Quantitative results of U-Net results and performance analysis of VarNet are also provided in the Appendix 8.

Additional findings on experiments show increased undersampling or new anatomies, modalities and datasets show that non-TTA often outperforms zero-filling, especially for U-Net. However, under large mask shifts, unrolled models like VarNet suffer more severe degradation without adaptation, underscoring the necessity of TTA in such cases.

**Qualitative Results.** Figure 5 present visual comparisons of reconstructed images for the FINE-based UNet methods in the anatomy shift. More results of other distribution shifts and VarnNet are provided in the Appendix. Self-supervised methods without MR-INR (e.g., FINE [38]) may risk over-smoothing when confronted with limited data, as highlighted in the error maps. While FINE+SST improves over FINE by incorporating single-slice adaptation, it lacks the AD module, leading to over-smoothing and loss of structural details. Our proposed approach, which integrates MR-INR with SST and AD, effectively balances adaptation and detail preservation, reducing hallucinations and enhancing reconstruction quality.

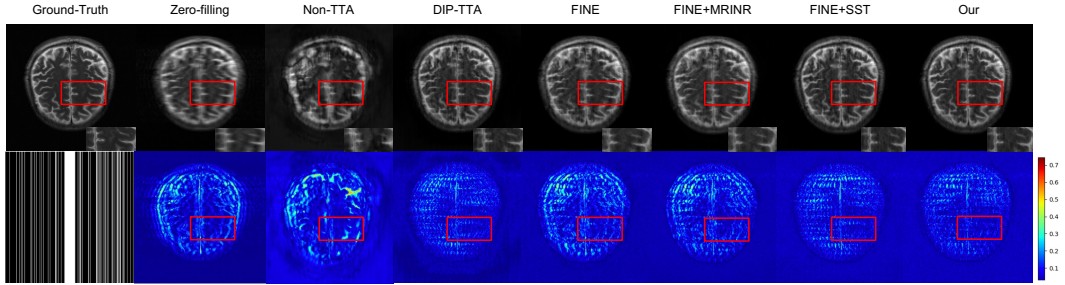

Figure 5: Comparison of different frameworks in UNet under anatomy shift (Knee to Brain) using the FINE method. The first row shows reconstructed MRI images, while the second row presents residual maps between reconstructions and full-sampled MRI. The proposed method (far right) achieves the lowest residuals, indicating improved reconstruction accuracy.

**Ablation Study.** We conduct a comprehensive ablation to disentangle the contributions of MR-INR, SST, and AD to adaptation performance under anatomy shift (Table 2 and Table 3). For both U-Net and VarNet backbones, introducing MR-INR on top of FINE yields clear PSNR improvements (e.g., +0.5 dB on U-Net), demonstrating that the lightweight latent code provides effective patient-specific adaptation with negligible parameter increase. Further integrating SST significantly boosts performance but comes with increased inference time, highlighting its complementary role in capturing slice-level variations. Importantly, incorporating AD alongside frozen CNN achieves the best trade-off between performance and efficiency (27.71 dB / 12.1 min on U-Net; 27.68 dB / 17.1 min on VarNet), outperforming both purely patient-wise or slice-wise training. This suggests that AD effectively compensates for the lack of full fine-tuning, particularly in constrained adaptation settings, and enhances the robustness of MR-INR+SST pipelines. Interestingly, while AD brings limited improvement when used with a fully trainable CNN, it yields notable gains when the CNN are frozen. This highlights AD's effectiveness in constrained settings, where its adaptive capacity compensates for the lack of end-to-end fine-tuning. This effect is further supported by the results in the last two rows of the table.

We conduct a sensitivity analysis on the AD step size (Table 4). We observe that reconstruction quality is stable across a wide range of step sizes, with the best PSNR/SSIM obtained at 1.0 and only marginal degradation at smaller values. Meanwhile, LPIPS improves slightly as the step size decreases, indicating a tunable trade-off between fidelity and perceptual similarity. This robustness suggests that AD is insensitive to moderate hyperparameter variations, which is desirable for test-time deployment. Similarly, Figure 6 shows that adding directional and adaptive priors steadily improves PSNR and SSIM, further validating the effectiveness of the adaptive components.

Additional results on in-domain adaptation, statistical analysis, sensitivity analysis, and more visualizations are provided in Appendix 8.

## 7    Conclusion and Limitation

We presented D2SA framework, a test-time adaptation framework that improves MRI reconstruction under distribution shifts. D2SA combines patient-wise MR-INR for modeling mean/variance shifts and single-slice refinement via a learnable AD module. This dual-stage design enhances generalisation, preserves structural fidelity, and accelerates convergence. Extensive experiments across five domain shifts demonstrate that D2SA consistently outperforms prior TTA approaches in both quality and efficiency. Ablation studies further validate the contributions of MR-INR, AD, and frozen layers in balancing performance and runtime. While D2SA reduces adaptation time and improves generalisation, several limitations remain. First, current evaluations are limited to publicly available datasets; further validation on real-world clinical undersampled MRI is necessary. Second, the framework operates on 2D slices independently—extending it to exploit full 3D spatial correlations is a natural next step. Finally, integrating stronger priors (e.g., diffusion models) and developing online or incremental learning strategies could further enhance adaptability and prevent forgetting when adapting to continuous patient streams.

| Ablation (UNet) | PSNR ↑ / Params ↓ / Time ↓ |
|---|---|
| *Patient-wise training* | |
| FINE | 25.98 / 31.02 / **4.9** |
| +MR-INR (❄ latent code) | 26.37 / 31.29 / 5.3 |
| +MR-INR (🔥 latent code) | **26.48** / **31.29** / 5.5 |
| *Single-slice training without stage 1* | |
| +SST (❄ cnn + 🔥 AD) | 27.31 / 17.62 / 18.7 |
| *Single-slice training with stage 1* | |
| +SST (🔥 cnn) | 27.22 / 31.02 / 17.2 |
| +SST (❄ cnn + 🔥 AD) | 27.37 / 17.62 / 22.6 |
| +MR-INR+SST (🔥 cnn) | 27.65 / 31.29 / 23.65 |
| +MR-INR+SST (🔥 cnn+🔥 AD) | 27.54 / 46.13 / 15.7 |
| +MR-INR❄+SST(❄cnn+🔥AD) | 27.41 / 17.62 / 14.5 |
| +MR-INR+SST(❄ cnn) | 27.38 / **3.05** / 17.5 |
| +MR-INR+SST(❄cnn+🔥AD)(Ours) | **27.71** / 17.89 / **12.1** |

| Ablation (VarNet) | PSNR ↑ / Params ↓ / Time ↓ |
|---|---|
| *Patient-wise training* | |
| FINE | 24.01 / 29.45 / **3.9** |
| +MR-INR (❄ latent code) | 26.37 / 29.69 / 5.3 |
| +MR-INR (🔥 latent code) | **26.45** / **29.69** / 4.7 |
| *Single-slice training without stage 1* | |
| +SST (❄ cnn + 🔥 AD) | 27.50 / 16.74 / 56.8 |
| *Single-slice training with stage 1* | |
| +SST (🔥 cnn) | 27.57 / 29.45 / 53.6 |
| +SST (❄ cnn + 🔥 AD) | 27.58 / 16.74 / 53.3 |
| +MR-INR+SST (🔥 cnn) | 27.68 / 29.69 / 20.9 |
| +MR-INR+SST (🔥 cnn+🔥 AD) | 27.61 / 43.79 / 34.0 |
| +MR-INR❄+SST(❄cnn+🔥AD) | 27.63 / 16.74 / 27.8 |
| +MR-INR+SST(❄ cnn) | 27.45 / **2.88** / 21.6 |
| +MR-INR+SST(❄cnn+🔥AD)(Ours) | **27.68** / 16.98 / **17.1** |

Table 2: Ablation study on MR-INR and AD under anatomy shift (U-Net). Each row reports PSNR ↑, parameter count (Millions) ↓, and inference time (min/patient) ↓. The latent codes only have 1408 parameters in this shift. Stage 1 compares MR-INR variants; Stage 2 evaluates SST with and without frozen MR-INR and AD.

Table 3: Ablation study on MR-INR and AD under anatomy shift (VarNet). Each row reports PSNR ↑, parameter count (Millions) ↓, and inference time (min/patient) ↓. The latent codes only have 1408 parameters in this shift. Stage 1 compares MR-INR variants; Stage 2 evaluates SST with and without frozen MR-INR and AD.

| AD Step Size | PSNR ↑ | SSIM ↑ | LPIPS ↓ |
|---|---|---|---|
| 1.0 | 27.71 | 0.876 | 0.320 |
| 0.1 | 27.38 | 0.874 | 0.322 |
| 0.01 | 27.36 | 0.874 | 0.323 |

Table 4: Sensitivity analysis of AD step size. Reconstruction quality remains stable across step sizes, with the best PSNR/SSIM at 1.0 and slightly improved LPIPS for smaller values, indicating a tunable fidelity–perception trade-off and robustness to hyperparameter variations.

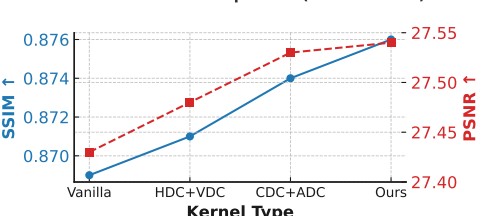

Figure 6: Ablation study on different kernel designs in difference convolution in terms of SSIM ↑ and PSNR ↑. Progressive improvements are observed from Vanilla to HD+VD, CD+AD, and ours, demonstrating the effectiveness of direction-aware and adaptive designs.

# 8 Acknowledgement

LZ gratefully acknowledges the financial aid award from NeurIPS and the travel support from DAMTP and Queens' College. RS acknowledges the support from Future Network of Intelligence Institute, The Chinese University of Hong Kong (Shenzhen). ZD acknowledges the support from Wellcome Trust 221633/Z/20/Z and the funding from the Cambridge Centre for Data-Driven Discovery and Accelerate Program for Scientific Discovery, made possible by a donation from Schmidt Sciences. YC is funded by an AstraZeneca studentship and a Google studentship. CBS acknowledges support from the Philip Leverhulme Prize, the Royal Society Wolfson Fellowship, the EPSRC advanced career fellowship EP/V029428/1, EPSRC grants EP/S026045/1 and EP/T003553/1, EP/N014588/1, EP/T017961/1, the Wellcome Innovator Awards 215733/Z/19/Z and 221633/Z/20/Z, the European Union Horizon 2020 research and innovation programme under the Marie Skodowska-Curie grant agreement No.777826 NoMADS, the Cantab Capital Institute for the Mathematics of Information and the Alan Turing Institute. AIAR gratefully acknowledges the support from Yau Mathematical Sciences Center, Tsinghua University.

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

# D2SA: Dual-Stage Distribution and Slice Adaptation for Efficient Test-Time Adaptation in MRI Reconstruction- Appendix

## Contents

# A   Training Details

This section outlines the key configurations, optimisation strategies, and architectural choices employed during training.

**Stage 1: Functional-Level Patient Adaptation.** We train the MR-INR-based model using a batch size of 2. Two Adam optimizers are used: one with a learning rate of $10^{-4}$ for the MR-INR weights and the base network, and the other with a learning rate of $10^{-3}$ for the learnable latent code. The training runs for 25 epochs, with convergence typically observed after 20 epochs. The 1D latent code (size $1 \times 128$) is initialized using a zero-mean multivariate Gaussian distribution with standard deviation $\sigma = 0.01$.

For the SIREN architecture [29], we use a 4-layer MLP with 256 hidden units per layer and follow the weight initialization method from the original paper. In U-Net, the loss coefficients in this stage are set to: $\lambda_{\text{self}} = 1$, $\lambda_{\text{INR}} = 1$, and $\lambda_{\text{reg}} = 1\mathrm{e}^{-4}$. For VarNet, we use $\lambda_{\text{self}} = 1$, $\lambda_{\text{INR}} = 1\mathrm{e}^{-3}$, and $\lambda_{\text{reg}} = 1\mathrm{e}^{-4}$.

**Stage 2: Single-Slice Refinement.** In this stage, we refine each slice using a learnable Anisotropic Diffusion (AD) module while keeping the original convolutional layers frozen. We use the Adam optimizer with a learning rate of $10^{-4}$ and train for up to 1000 steps.

Following the self-validation strategy in [11], we reserve 5% of the k-space signals for validation. If the validation error does not decrease in fixed iterations, the refinement process is terminated early. For early stopping, we apply a sliding window of size 30 to monitor the moving average of the validation error for methods including FINE, NR2N, and SSDU (with and without MR-INR). DIP-TTT uses a sliding window size of 100, as defined in the original repository [11]. The loss coefficients for U-Net are $\lambda_{\text{self}} = 1$, $\lambda_{\text{INR}} = 1$, and for VarNet, they are $\lambda_{\text{self}} = 1$, $\lambda_{\text{INR}} = 1\mathrm{e}^{-3}$.

**Mask Setup.** First, a general under-sampling mask is set by using 1D Cartesian masks with an acceleration rate of $\times 4$ and 8% auto-calibrating lines.

- For *Anatomy*, *Dataset*, and *Modality* shifts, the same $\times 4$ random 1D Cartesian mask is applied to both source and target domains. - For the *Acceleration* shift, the source model is trained on MR signals under-sampled at $\times 2$ using the same 8% auto-calibration strategy, while the target domain uses a $\times 4$ mask with the same random seed to simulate the shift. - For the *Sampling* shift, the source domain uses a random $\times 4$ mask, and the target domain uses a uniform $\times 4$ mask, both with 8% calibration lines. - Specifically, for in-distribution testing under sampling shift, we apply different random seeds to generate the masks while keeping the sampling strategy ($\times 4$, random) unchanged.

All mask generation and TTA implementation are provided in the demo script.

# B   Supplementary Quantitative Results in OOD Shift

**Performance comparison of UNet methods under different domain shifts.** Table 5 reports comprehensive and direct quantitative comparisons of UNet-based reconstruction methods under five distinct types of domain shift: anatomy, dataset, modality, acceleration, and sampling. Our proposed two-stage strategy combines patient-level adaptation (MR-INR) and slice-level refinement (SST with AD module). It consistently achieves top or near-top results across SSIM, PSNR, and LPIPS. Meanwhile, the inference time remains practical and competitive. Notably, the two-stage models yield especially strong improvements under more challenging shifts such as dataset and modality, where domain discrepancies are typically larger. These gains demonstrate the benefit of globally shared representations learned during patient-level adaptation, which are then effectively refined with localized slice-level refinemennt training. Furthermore, compared to strong baselines like DIP-TTT and one-stage TTA methods (e.g., +SST), our models exhibit improved stability (lower LPIPS) and higher sample-level fidelity (SSIM/PSNR), highlighting the robustness and generalization capacity of our hierarchical test-time learning approach under OOD scenarios.

**Visual Analysis of VarNet Performance Across Domain Shifts.** Figure 7 provides a detailed visualization of VarNet's performance under various domain shifts based on the Table of VarNet in main paper. The bar chart (top-left) highlights PSNR gains achieved by incorporating MR-INR in Stage 1 across FINE, NR2N, and SSDU training strategies. Acceleration and sampling shifts benefit the most, reflecting MR-INR's ability to capture cross-slice anatomical consistency in challenging

| Method (UNet) | Anatomy Shift ($\mathcal{S}$: Knee, $\mathcal{T}$: Brain) | Dataset Shift ($\mathcal{S}$: Stanford, $\mathcal{T}$: fastMRI) | Modality Shift ($\mathcal{S}$: AXT2, $\mathcal{T}$: AXT1PRE) | Acceleration Shift ($\mathcal{S}$: 2x, $\mathcal{T}$: 4x) | Sampling Shift ($\mathcal{S}$: Random, $\mathcal{T}$: Uniform) |
|---|---|---|---|---|---|
| Zero-filling | 0.737/24.50/0.327/- | 0.754/24.33/0.359/- | 0.747/25.7/0.350/- | 0.754/23.371/0.396/- | 0.765/26.28/0.338/- |
| Non-TTA | 0.625/21.77/0.458/- | 0.559/21.87/0.454/- | 0.794/27.18/0.391/- | 0.726/23.37/0.396/- | 0.825/26.97/0.376/- |
| DIP-TTT | 0.859/27.05/0.322/42.1 | 0.810/28.08/0.298/40.8 | 0.846/27.61/0.361/31.5 | 0.815/27.93/0.299/95.3 | 0.894/28.98/0.314/27.1 |
| FINE | 0.834/25.98/0.351/4.9 | 0.796/26.54/0.319/6.4 | 0.825/26.71/0.377/5.6 | 0.782/25.75/0.333/6.6 | 0.872/27.99/0.334/5.1 |
| FINE+MR-INR | 0.845/26.37/0.346/5.5 | 0.807/26.84/0.314/6.6 | 0.835/26.51/0.373/6.0 | 0.793/26.29/0.326/7.0 | 0.876/28.24/0.333/5.2 |
| FINE+SST | 0.868/27.22/0.327/17.2 | 0.827/28.16/0.283/21.9 | 0.853/27.72/0.293/21.9 | 0.822/28.07/0.389/52.2 | 0.891/29.02/0.314/18.4 |
| FINE+MR-INR+SST | 0.876/27.71/0.320/12.1 | 0.829/28.34/0.279/18.7 | 0.861/27.93/0.279/15.7 | 0.825/28.54/0.286/31.9 | 0.895/29.05/0.311/13.2 |
| NR2N | 0.836/25.80/0.353/5.2 | 0.796/26.71/0.316/6.9 | 0.826/26.59/0.383/6.7 | 0.781/26.20/0.335/6.9 | 0.859/26.91/0.347/5.6 |
| NR2N+MR-INR | 0.849/26.11/0.346/5.7 | 0.798/26.42/0.317/7.4 | 0.829/26.64/0.380/7.3 | 0.791/26.37/0.332/7.5 | 0.867/27.43/0.343/5.7 |
| NR2N+SST | 0.868/27.38/0.323/21.7 | 0.825/28.23/0.284/22.5 | 0.854/27.69/0.284/22.6 | 0.822/28.07/0.291/52.7 | 0.892/29.08/0.313/19.3 |
| NR2N+MR-INR+SST | 0.871/27.32/0.323/12.2 | 0.830/28.43/0.279/16.4 | 0.862/27.98/0.279/14.5 | 0.825/28.86/0.287/31.7 | 0.895/29.12/0.311/12.2 |
| SSDU | 0.851/24.82/0.353/5.4 | 0.788/22.37/0.344/7.8 | 0.819/24.27/0.339/7.1 | 0.789/23.03/0.346/7.3 | 0.856/24.88/0.349/6.5 |
| SSDU+MR-INR | 0.861/25.18/0.348/5.6 | 0.789/22.45/0.339/8.0 | 0.832/24.97/0.385/7.4 | 0.797/23.78/0.344/7.7 | 0.873/26.11/0.347/6.6 |
| SSDU+SST | 0.871/25.17/0.349/25.3 | 0.825/28.35/0.284/30.6 | 0.854/27.71/0.287/25.7 | 0.823/28.07/0.293/139.8 | 0.893/29.02/0.315/28.1 |
| SSDU+MR-INR+SST | 0.877/27.46/0.322/11.5 | 0.828/28.36/0.287/18.9 | 0.860/28.04/0.287/17.4 | 0.826/28.62/0.286/44.2 | 0.897/29.04/0.310/14.6 |

Table 5: Performance comparison of UNet methods under different domain shifts. Each cell presents (**SSIM** ↑ / **PSNR** ↑ / **LPIPS** ↓ / **Time (mins/patient)** ↓). The proposed methods combine MR-INR-based patient-wise adaptation and single-slice refinement.

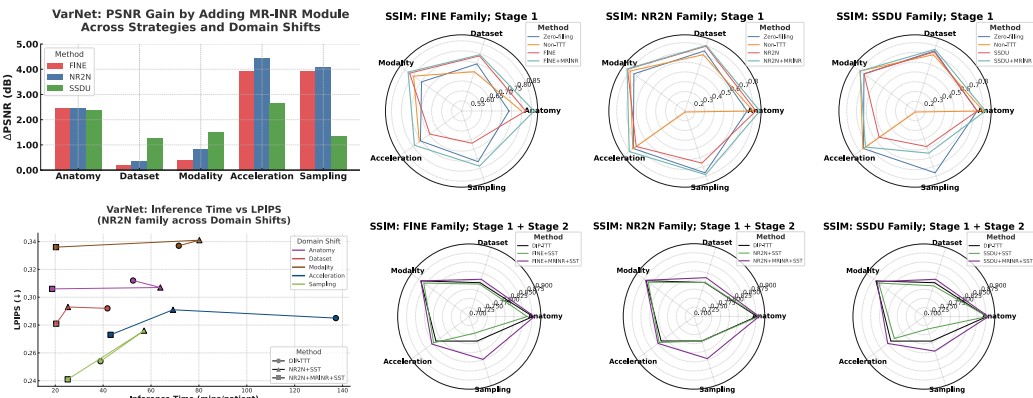

Figure 7: Performance Analysis of VarNet under Different Domain Shifts. Top-left: PSNR gain from adding MR-INR across FINE, NR2N, and SSDU in stage 1, with the largest gain in modality shift in most strategies. Bottom-left: LPIPS vs. inference time trade-off showing that +MR-INR+SST achieves higher quality with reduced time comsumption compared to DIP-TTT and normal +SST. Right: SSIM across five domain shifts for each SSL family, visualized patient-level (Stage 1) and after slice refinement (Stage 1 + Stage 2). MR-INR consistently improves performance across domains, and combining it with SST further enhances SSIM.

contexts. The LPIPS vs. inference time plot (bottom-left) illustrates that MR-INR+SST strikes a favorable balance between reconstruction quality and computational efficiency, outperforming DIP-TTT and conventional SST in both metrics. Radar plots (right) further validate our hierarchical design: Stage 1 improves SSIM through patient-level adaptation, while Stage 2 refinement with SST delivers additional performance boosts. These results reflect trends observed in Figure of performance analysis on UNet and confirm that our framework generalizes effectively to VarNet, enhancing robustness and generalization under all OOD shifts.

**Distributional Analysis.** To further assess the robustness of our method, Figure 8 presents violin plots of adjusted SSIM, PSNR, and LPIPS metrics under two representative domain shifts: *acceleration* (top row, Knee dataset with UNet) and *sampling* (bottom row, Brain dataset with VarNet). $\mu$ and $\sigma$ denote the mean and standard deviation of each metric across slices per subject. Adjusted values are computed as $\mu - 0.5\sigma$ per subject, considering both and emphasizing performance stability across slices per patient. Notably, SSDU+MR-INR+SST consistently outperforms both DIP-TTT and SSDU+SST across all metrics. Improvements are statistically significant in most cases, particularly for PSNR and LPIPS ($p < 0.01$, Wilcoxon signed-rank test), highlighting our framework's ability to deliver high-fidelity and stable reconstructions. These results reaffirm that the two-stage strategy

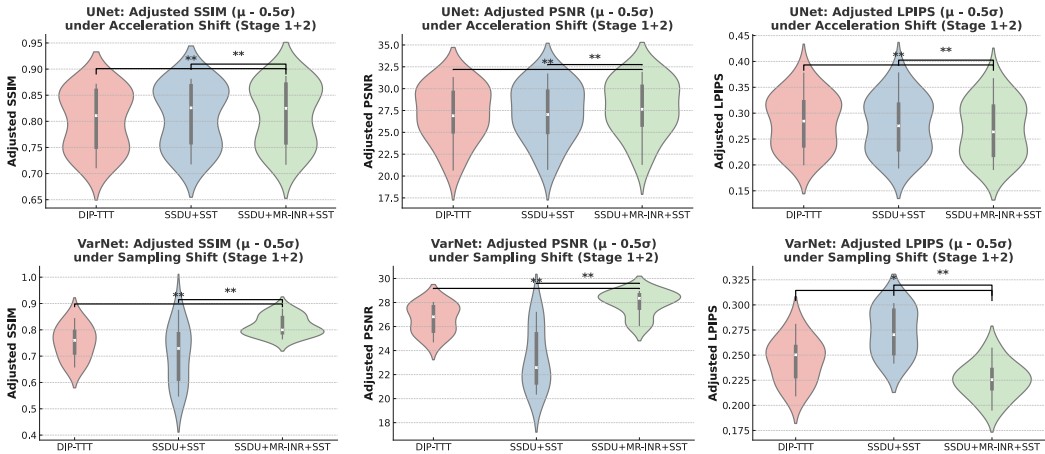

Figure 8: Comparison of model performance under domain shifts in **acceleration** (top) and **sampling** (bottom). Violin plots show the per-subject distributions of adjusted SSIM ($\uparrow$), adjusted PSNR ($\uparrow$), and adjusted LPIPS ($\downarrow$), where each adjusted metric is computed as $\mu - 0.5\sigma$, with $\mu$ and $\sigma$ denoting the mean and standard deviation of each metric across slices per subject. Three test-time strategies are evaluated: DIP-TTT, SSDU+SST, and SSDU+MR-INR+SST. The top row presents results under acceleration shift on the **Knee dataset** using UNet, while the bottom row corresponds to sampling shift on the **Brain dataset** using VarNet. Mean values are marked within each violin. Statistical significance of SSDU+MR-INR+SST compared to baseline methods (DIP-TTT and SSDU+SST) is indicated by $^{*}$ ($p < 0.05$) or $^{**}$ ($p < 0.01$), based on Wilcoxon signed-rank tests.

(+MR-INR+SST with AD module) offers superior stability under diverse and challenging OOD shifts.

## C Comparison to ZS-SSL

Recent progress in TTA MRI reconstruction has introduced zero-shot paradigms such as ZS-SSL [35], which train directly on undersampled measurements from a single subject. While effective in few data cases, ZS-SSL reconstructs each slice independently, partitioning and augmenting available k-space into training, loss, and validation sets. This design can limit the model's ability to capture shared anatomical representation.

Another representative method, DIP-TTT [11], improves upon the original Deep Image Prior by incorporating early stopping to stabilize optimization during test-time training. However, similar to ZS-SSL, it operates slice-by-slice and lacks modeling of inter-slice spatial dependencies. While DIP-TTT has been shown to outperform ZS-SSL under anatomy shifts, broader comparisons across other domain shifts are missing.

In contrast, our proposed approach adopts a two-stage test-time adaptation strategy: (1) MR-INR performs subject-level adaptation by leveraging all patient scans to capture shared anatomical structure and slice-wise relationships, and (2) SST refines each slice independently via AD module. This hierarchical modeling allows us to exploit both global context and local detail refinement during inference.

Table 6 presents quantitative results under five domain shifts: anatomy, dataset, modality, acceleration, and sampling. ZS-SSL shows competitive performance under anatomy and modality shifts, but it does not achieve surpassing on all metrics and spends more time to converge. It lags behind our methods in all other settings. Our MR-INR+SST framework consistently achieves state-of-the-art results across metrics and shifts. Furthermore, it significantly reduces inference time (e.g., 17–25 min vs. 100+ min for ZS-SSL), highlighting its practical applicability. More details about the the comparison on visualisation are in the last section-Supplementary Visualisations.

These findings suggest that the hierarchical design of our two-stage adaptation is better suited to handling diverse and challenging distribution shifts than methods relying solely on single-slice

reconstruction. By explicitly modeling cross-slice dependencies in Stage 1 and adapting locally in Stage 2, our method achieves strong generalization and efficiency in real-world deployment scenarios.

| Method | Anatomy Shift ($\mathcal{S}$: Knee, $\mathcal{T}$: Brain) | Dataset Shift ($\mathcal{S}$: Stanford, $\mathcal{T}$: fastMRI) | Modality Shift ($\mathcal{S}$: AXT2, $\mathcal{T}$: AXT1PRE) | Acceleration Shift ($\mathcal{S}$: 2x, $\mathcal{T}$: 4x) | Sampling Shift ($\mathcal{S}$: Random, $\mathcal{T}$: Uniform) |
|---|---|---|---|---|---|
| Zero-filling | 0.737/24.50/0.327/- | 0.747/24.33/0.359/- | 0.747/25.71/0.350/- | 0.754/23.37/0.396/- | 0.766/26.28/0.338/- |
| Non-TTT | 0.799/23.16/0.371/- | 0.706/22.35/0.365/- | 0.796/23.54/0.379/- | 0.761/23.04/0.372/- | 0.111/16.00/0.594/- |
| DIP-TTT | 0.878/27.67/0.312/52.5 | 0.798/28.02/0.292/41.8 | 0.867/28.33/0.337/71.6 | 0.815/28.25/0.285/137.2 | 0.771/27.65/0.254/38.9 |
| ZS-SSL | 0.884/27.07/0.314/99.5 | 0.745/21.93/0.365/135.4 | 0.860/28.31/0.338/103.3 | 0.751/21.37/0.381/167.4 | 0.734/24.30/0.306/52.2 |
| FINE+SST | 0.862/27.57/0.311/53.5 | 0.794/27.72/0.294/20.3 | 0.857/28.08/0.345/79.8 | 0.823/28.17/0.288/63.8 | 0.748/25.18/0.276/48.3 |
| NR2N+SST | 0.883/27.72/0.307/63.9 | 0.798/27.78/0.293/25.3 | 0.860/28.17/0.341/80.2 | 0.822/28.09/0.291/69.2 | 0.771/26.46/0.276/57.1 |
| SSDU+SST | 0.879/27.65/0.310/68.3 | 0.789/26.79/0.299/24.9 | 0.857/28.07/0.343/93.4 | 0.803/28.06/0.293/134.2 | 0.736/24.84/0.288/50.5 |
| FINE+MRINR+SST | 0.882/27.68/0.311/17.1 | 0.808/28.72/0.286/18.2 | 0.867/28.32/0.337/21.8 | 0.829/28.64/0.276/44.5 | 0.824/28.49/0.232/22.1 |
| NR2N+MRINR+SST | 0.884/27.81/0.306/18.7 | 0.812/28.76/0.281/20.4 | 0.869/28.36/0.336/20.3 | 0.826/28.89/0.273/43.1 | 0.822/28.40/0.241/25.2 |
| SSDU+MRINR+SST | 0.882/27.66/0.307/18.5 | 0.808/28.16/0.290/21.4 | 0.863/28.09/0.342/29.3 | 0.826/28.62/0.286/45.2 | 0.800/28.21/0.254/25.8 |

Table 6: Cross-domain evaluation under five domain shifts: anatomy, dataset, modality, acceleration, and sampling. Each cell shows the model performance in format **SSIM** (↑) / **PSNR** (↑) / **LPIPS** (↓) / **Time (mins/patient)** (↓). Rows shaded in light red highlight our proposed MR-INR+SST methods, which achieve consistent improvements across shifts with notably reduced inference time.

# D  Other Ablation Studies and Sensitivity Analysis

**Choice of INRs**   To clarify our choice of SIREN as the INR backbone in MR-INR, we conducted an additional ablation comparing different INR backbones under both patient-wise (FINE+MR-INR) and single-slice (FINE+MR-INR+SST) training. Results are summarized below:

| Backbone (Stage 1) | SSIM↑ / PSNR↑ / LPIPS↓ | Backbone (Stage 1+2) | SSIM↑ / PSNR↑ / LPIPS↓ |
|---|---|---|---|
| WIRE | 0.839 / 25.78 / 0.354 | Finer | 0.873 / 27.40 / 0.323 |
| Finer | 0.845 / 26.05 / 0.350 | WIRE | 0.872 / 27.44 / 0.321 |
| RPE + MLP | 0.845 / 25.76 / 0.349 | RPE + MLP | 0.864 / 27.22 / 0.326 |
| SIREN only | 0.845 / 26.16 / 0.349 | SIREN only | 0.874 / 27.51 / 0.322 |
| **Ours (PE+SIREN)** | **0.845 / 26.37 / 0.346** | **Ours (PE+SIREN)** | **0.876 / 27.71 / 0.320** |

Table 7: Sensitivity of INR backbone choice under patient-wise (Stage 1) and single-slice (Stage 1+2) Refinement.

Our design combining random Fourier positional encoding with SIREN consistently yields the best or near-best performance, especially in perceptual quality (LPIPS). While Finer offers frequency adaptability, its more complex architecture increases overhead without clear performance gains; WIRE, though efficient, suffers from unstable convergence and limited representation in medical shift settings. Compared to INCODE, our SIREN-based MR-INR avoids the need for pretraining, supporting our lightweight, plug-and-play adaptation objective.

**Loss Weight Ablation**   We further evaluate the sensitivity of the framework to different loss weight configurations for Stage 1 ($\lambda_{\text{Self}}, \lambda_{\text{INR}}, \lambda_{\text{Reg}}$) and Stage 2 ($\lambda_{\text{Self}}, \lambda_{\text{INR}}$) (Table 8). Overall performance remains stable across a wide range of $\lambda$ values, demonstrating the robustness of the training objectives. However, setting $\lambda_{\text{INR}}$ too low (e.g., $0.1$) leads to notable degradation, especially in Stage 1 and LPIPS in Stage 2. This aligns with the key role of MR-INR in capturing patient-level distribution shifts, where a sufficient INR loss weight is critical for effective adaptation.

| Stage | $\lambda$ Settings | PSNR ↑ | SSIM ↑ | LPIPS ↓ |
|---|---|---|---|---|
| Stage 1 | 1 / 1 / 1e-4 | 24.97 | 0.832 | 0.385 |
|  | 1 / 0.1 / 1e-4 | 23.85 | 0.823 | 0.386 |
|  | 1 / 1 / 0.1 | 24.00 | 0.822 | 0.388 |
| Stage 2 | 1 / 1 | 28.04 | 0.860 | 0.287 |
|  | 1 / 0.1 | 28.00 | 0.861 | 0.356 |
|  | 1 / 0.01 | 27.99 | 0.861 | 0.350 |

Table 8: Ablation of loss weight combinations. Performance is stable across a wide range of $\lambda$ values, but mild degradation when $\lambda_{\text{INR}}$ is too small.

**Hyper-parameter Sensitivity.** We evaluate the sensitivity of MR-INR (Stage 1) and SST+AD (Stage 2) to key hyperparameters, including latent code size, INR depth, and initialization scale (Table 9). The performance across PSNR, SSIM, and LPIPS remains remarkably stable, with less than 0.15 dB variation in PSNR and negligible LPIPS differences. This robustness indicates that the method is largely insensitive to moderate hyperparameter changes, making it well suited for deployment without extensive tuning.

| TTA Stage | Hyperparameter | SSIM ↑ | PSNR ↑ | LPIPS ↓ |
|---|---|---|---|---|
| | Latent Code Size (64 / 128 / 256) | 0.849 / 0.845 / 0.850 | 26.39 / 26.37 / 26.47 | 0.345 / 0.346 / 0.335 |
| Stage 1 | INR Depth (3 / 4 / 5) | 0.842 / 0.845 / 0.848 | 26.34 / 26.37 / 26.44 | 0.345 / 0.346 / 0.346 |
| | Init Std (0.1 / 0.01 / 0.001) | 0.845 / 0.845 / 0.847 | 26.50 / 26.37 / 26.37 | 0.346 / 0.346 / 0.345 |
| | Latent Code Size (64 / 128 / 256) | 0.876 / 0.876 / 0.877 | 27.53 / 27.71 / 27.53 | 0.323 / 0.320 / 0.320 |
| Stage 2 | INR Depth (3 / 4 / 5) | 0.874 / 0.876 / 0.877 | 27.54 / 27.71 / 27.56 | 0.321 / 0.320 / 0.319 |
| | Init Std (0.1 / 0.01 / 0.001) | 0.876 / 0.876 / 0.875 | 27.53 / 27.71 / 27.52 | 0.321 / 0.320 / 0.322 |

Table 9: Hyperparameter sensitivity of MR-INR (Stage 1) and SST+AD (Stage 2). The method remains stable under variations in latent code size, network depth, and initialization, with PSNR fluctuations < 0.15 dB and negligible LPIPS differences.

# E   Quantitative Results in Same Domain Shift

**In-domain Generalization Analysis.** Table 10 reports the performance of UNet-based reconstruction methods evaluated under same-domain (in-domain) settings across five shifts: *Brain*, *fastMRI*, *AXT1PRE*, *2x*, and *Random*. As expected, most methods perform better under in-domain testing compared to out-of-distribution (OOD) settings, with consistently higher SSIM and PSNR and lower LPIPS scores. Among the baselines, DIP-TTT demonstrates strong performance, especially in settings like Brain and AXT1PRE. However, it comes at a significantly higher inference cost, as reflected in its per-patient runtime.

Notably, our proposed two-stage pipeline (MR-INR+SST) achieves the best or second-best scores across almost all shifts, outperforming both FINE+SST and DIP-TTT in both fidelity and efficiency. For instance, in the sampling in same domain shift, SSDU+MR-INR+SST improves SSIM and PSNR while maintaining reduced LPIPS and halving the inference time compared to DIP-TTT. Furthermore, combining patient-level modeling (MR-INR) with slice-level SST refinement results in consistent performance gains over single-stage variants, demonstrating the benefit of hierarchical adaptation even in-domain.

This table highlights the robustness and generality of our framework: even when domain shifts are minimal in some cases, dual-stage adaptation continues to yield meaningful improvements in both reconstruction quality and computational efficiency.

| Method (UNet) | Brain → Brain | fastMRI → fastMRI | AXT1PRE → AXT1PRE | 2x → 2x | Random → Random |
|---|---|---|---|---|---|
| Zero-filling | 0.737/24.50/0.359/- | 0.754/24.33/0.359/- | 0.747/25.70/0.350/- | 0.846/26.52/0.226/- | 0.811/26.32/0.387 |
| Non-TTT | 0.822/26.50/0.358/- | 0.559/21.88/0.454/- | 0.799/26.08/0.395/- | 0.149/15.74/0.580/- | 0.764/25.87/0.331 |
| DIP-TTT | 0.876/27.45/0.323/46.1 | 0.806/28.43/0.281/62.9 | 0.858/27.87/0.354/15.1 | 0.834/28.63/0.207/95.4 | 0.870/28.18/0.342/22.3 |
| FINE | 0.847/26.39/0.345/4.6 | 0.799/26.88/0.309/6.9 | 0.837/26.88/0.368/4.5 | 0.846/26.07/0.275/7.1 | 0.853/27.32/0.357/4.7 |
| FINE+MRINR | 0.852/26.66/0.343/13.5 | 0.805/27.08/0.312/7.2 | 0.839/27.28/0.366/4.7 | 0.878/28.40/0.229/7.4 | 0.856/27.44/0.356/4.9 |
| FINE+SST | 0.874/27.36/0.327/17.0 | 0.824/28.14/0.285/33.2 | 0.860/27.82/0.285/11.3 | 0.893/30.02/0.675/59.1 | 0.864/28.27/0.343/14.3 |
| FINE+MRINR+SST | 0.878/27.59/0.320/13.5 | 0.825/28.44/0.287/29.1 | 0.862/28.06/0.280/9.3 | 0.901/30.15/0.195/39.3 | 0.874/28.29/0.335/13.2 |
| NR2N | 0.850/26.11/0.347/4.8 | 0.799/26.92/0.307/7.3 | 0.835/26.88/0.374/4.7 | 0.836/25.09/0.286/7.2 | 0.850/27.40/0.362/4.8 |
| NR2N+MRINR | 0.857/26.18/0.345/5.0 | 0.803/26.99/0.298/7.6 | 0.836/26.95/0.378/5.1 | 0.870/25.56/0.242/7.6 | 0.854/27.35/0.358/5.0 |
| NR2N+SST | 0.875/27.49/0.323/20.7 | 0.825/28.20/0.283/33.5 | 0.865/27.88/0.283/12.6 | 0.892/30.03/0.207/60.0 | 0.864/28.27/0.343/18.1 |
| NR2N+MRINR+SST | 0.876/27.46/0.322/13.1 | 0.826/28.35/0.281/30.4 | 0.866/28.16/0.281/11.2 | 0.900/30.07/0.196/39.3 | 0.872/28.26/0.336/12.3 |
| SSDU | 0.861/25.16/0.347/5.0 | 0.794/22.64/0.332/7.5 | 0.848/25.40/0.375/5.2 | 0.804/20.73/0.311/7.4 | 0.836/24.30/0.374/5.0 |
| SSDU+MRINR | 0.865/25.36/0.323/5.3 | 0.8018/22.71/0.335/7.7 | 0.826/24.06/0.335/5.6 | 0.833/21.41/0.279/7.7 | 0.853/25.53/0.370/5.1 |
| SSDU+SST | 0.876/27.39/0.323/24.1 | 0.823/28.13/0.291/42.7 | 0.860/27.58/0.291/13.4 | 0.741/24.71/0.432/90.2 | 0.869/28.24/0.343/23.2 |
| SSDU+MRINR+SST | 0.879/27.57/0.322/11.5 | 0.825/28.17/0.285/35.4 | 0.863/28.14/0.285/11.2 | 0.901/30.25/0.192/44.2 | 0.877/28.30/0.333/14.4 |

Table 10: Performance comparison of **UNet** methods under in-domain evaluation. Each cell reports **SSIM** (↑) / **PSNR** (↑) / **LPIPS** (↓) / **Time (mins/patient)** (↓). Shaded rows indicate methods that include the proposed MR-INR-based adaptation ( MRINR ) and further enhancement via slice-wise SST refinement ( MRINR+SST ).

**In-domain Evaluation on VarNet.** Table 11 presents a comprehensive comparison of VarNet-based reconstruction methods evaluated under in-domain conditions (including DIP-TTT and ZS-SSL). As expected, most methods demonstrate improved performance. Our proposed two-stage adaptation pipeline, incorporating MR-INR and SST, achieves strong results across most datasets in terms of SSIM, PSNR, and LPIPS.

Notably, our method (e.g., +MRINR+SST) attains the best SSIM, PSNR or LPIPS in four out of five shifts. An exception occurs in the AXT1PRE → AXT1PRE setting, where baseline+SST achieves slightly better metrics. In contrast, our dual-stage approach imposes a strong global anatomical prior, which promotes per-slice refinement in this homogeneous setting. A stronger global anatomical prior from MR-INR may slightly limit the flexibility of slice-level adaptation in this shift.

Despite this, our method still delivers highly competitive results with significantly less inference time compared to DIP-TTT. For instance, NR2N+MRINR+SST achieves a comparable PSNR of 28.27 in the T1 case within 18.4 minutes/patient, versus 25.8 minutes for DIP-TTT. This supports the claim that our framework achieves better trade-offs between quality and efficiency, making it favorable for real-world applications requiring fast implementation without sacrificing reconstruction accuracy.

| Method (VarNet) | Brain → Brain | fastMRI → fastMRI | AXT1PRE → AXT1PRE | 2x → 2x | Random → Random |
|---|---|---|---|---|---|
| Zero-filling | 0.754/24.33/0.359/- | 0.747/24.33/0.359/- | 0.747/25.70/0.350/- | 0.846/26.52/0.226/- | 0.764/25.88/0.331/- |
| Non-TTT | 0.845/24.60/0.305/- | 0.706/23.12/0.331/- | 0.838/22.48/0.354/- | 0.149/15.74/0.580/- | 0.111/15.98/0.593/- |
| DIP-TTT | 0.875/27.29/0.315/102.4 | 0.815/28.28/0.282/57.1 | 0.869/28.33/0.331/25.8 | 0.840/29.26/0.196/120.4 | 0.683/24.84/0.320/32.3 |
| ZS-SSL | 0.885/27.57/0.313/124.4 | 0.754/22.22/0.379/100.2 | 0.870/27.93/0.334/85.4 | 0.742/21.94/0.302/167.4 | 0.634/21.34/0.410/49.6 |
| FINE | 0.854/26.49/0.328/3.9 | 0.801/26.55/0.300/7.2 | 0.862/27.70/0.335/3.5 | 0.816/24.17/0.273/7.4 | 0.646/20.62/0.388/4.3 |
| FINE+MRINR | 0.857/26.66/0.325/4.7 | 0.804/26.80/0.297/7.8 | 0.855/27.46/0.343/4.2 | 0.857/26.62/0.225/7.8 | 0.767/24.45/0.351/4.4 |
| FINE+SST | 0.877/27.62/0.310/91.9 | 0.809/27.87/0.289/30.4 | 0.873/28.41/0.329/26.2 | 0.832/29.07/0.204/63.8 | 0.697/24.06/0.327/47.1 |
| FINE+MRINR+SST | 0.884/27.81/0.306/47.6 | 0.825/28.34/0.278/23.5 | 0.862/28.16/0.337/19.1 | 0.860/29.30/0.198/44.3 | 0.787/27.49/0.261/18.3 |
| NR2N | 0.868/26.82/0.321/4.4 | 0.815/26.95/0.285/7.8 | 0.871/27.43/0.332/4.3 | 0.805/23.37/0.285/7.6 | 0.640/20.76/0.388/4.5 |
| NR2N+MRINR | 0.868/26.97/0.319/5.1 | 0.808/26.48/0.290/8.1 | 0.859/27.14/0.340/4.9 | 0.835/25.71/0.248/8.0 | 0.768/24.56/0.348/4.6 |
| NR2N+SST | 0.880/27.69/0.307/110.3 | 0.812/27.77/0.288/33.7 | 0.878/28.53/0.323/26.8 | 0.838/29.05/0.199/59.3 | 0.696/24.13/0.328/48.4 |
| NR2N+MRINR+SST | 0.885/27.81/0.306/49.7 | 0.826/28.42/0.279/22.9 | 0.867/28.27/0.331/18.4 | 0.844/29.24/0.191/45.7 | 0.787/27.51/0.261/20.4 |
| SSDU | 0.838/24.69/0.344/4.7 | 0.686/19.12/0.370/8.2 | 0.825/22.81/0.376/4.8 | 0.597/17.44/0.366/8.1 | 0.521/17.72/0.421/5.3 |
| SSDU+MRINR | 0.839/24.89/0.342/5.1 | 0.713/19.72/0.350/8.7 | 0.809/22.18/0.369/5.3 | 0.695/19.02/0.318/8.5 | 0.583/19.00/0.401/5.4 |
| SSDU+SST | 0.881/27.57/0.310/127.2 | 0.802/27.77/0.289/40.7 | 0.871/28.31/0.331/30.4 | 0.819/26.07/0.243/70.5 | 0.677/23.02/0.340/41.3 |
| SSDU+MRINR+SST | 0.887/27.57/0.310/50.4 | 0.815/28.29/0.285/39.2 | 0.867/28.20/0.333/24.7 | 0.826/28.62/0.286/50.1 | 0.768/27.05/0.282/25.9 |

Table 11: Performance comparison of **VarNet** methods under in-domain evaluation settings. Each cell reports **SSIM** (↑) / **PSNR** (↑) / **LPIPS** (↓) / **Time (mins/patient)** (↓). Shaded rows indicate methods that include the proposed MR-INR-based adaptation ( MRINR ) and further enhancement via slice-wise SST refinement ( MRINR+SST ).

# F  Mathematical Analysis for Proposed Method

## F.1  Test Distribution and Problem Formulation

We consider a setting where the observed signal $\mathbf{y}$ is a corrupted version of the underlying clean signal $\mathbf{x}$, which follows the test distribution:

$$Q : \mathbf{y} = \mathbf{x} + \mathbf{z}, \quad \mathbf{x} = \mathbf{U}\mathbf{c} + \mu_Q, \quad \mathbf{c} \sim \mathcal{N}(0, I), \quad \mathbf{z} \sim \mathcal{N}(0, Is^2). \tag{12}$$

Here, $\mathbf{U} \in \mathbb{R}^{n \times d}$ is an orthonormal basis for the signal subspace, and $\mu_Q$ represents the mean shift in the test distribution. Our goal is to estimate $\mathbf{x}$ under this distribution shift.

## F.2  Self-Supervised Adaptation by Affine Transformation

To address the distribution shift from $P$ to $Q$, we introduce an adaptation mechanism that accounts for both variance and mean shifts. The optimal estimator for $\mathbf{x}$ under test-time training (TTT) is:

$$\hat{\mathbf{x}} = \alpha \mathbf{U}\mathbf{U}^T \mathbf{y} + \beta, \tag{13}$$

where $\alpha$ accounts for variance shifts, and $\beta$ corrects for mean shifts.

The self-supervised loss function is defined as:

$$L_{SS}(\alpha, \beta, \mathbf{U}, \mathbf{y}) = \mathbb{E}_Q \left[ \|\mathbf{y} - \alpha \mathbf{U}\mathbf{U}^T\mathbf{y} - \beta\|_2^2 \right] + \frac{2\alpha d}{n-d} \mathbb{E}_Q \left[ \|(\mathbf{I} - \mathbf{U}\mathbf{U}^T)\mathbf{y}\|_2^2 \right]. \quad (14)$$

**Expanding the First Term of $L_{SS}$:**

$$\mathbb{E}_Q \left[ \|\mathbf{y} - \alpha \mathbf{U}\mathbf{U}^T\mathbf{y} - \beta\|_2^2 \right] = \mathbb{E}_Q \left[ \mathbf{y}^T\mathbf{y} - 2\alpha\mathbf{y}^T\mathbf{U}\mathbf{U}^T\mathbf{y} - 2\beta^T\mathbf{y} \right.$$
$$\left. + \alpha^2\mathbf{y}^T\mathbf{U}\mathbf{U}^T\mathbf{U}\mathbf{U}^T\mathbf{y} + 2\alpha\beta^T\mathbf{U}\mathbf{U}^T\mathbf{y} + \beta^T\beta \right]. \quad (15)$$

Taking expectation:

$$\mathbb{E}_Q \left[ \mathbf{y}^T\mathbf{y} \right] - 2\alpha\mathbb{E}_Q \left[ \mathbf{y}^T\mathbf{U}\mathbf{U}^T\mathbf{y} \right] - 2\mathbb{E}_Q \left[ \beta^T\mathbf{y} \right]$$
$$+ \alpha^2\mathbb{E}_Q \left[ \mathbf{y}^T\mathbf{U}\mathbf{U}^T\mathbf{U}\mathbf{U}^T\mathbf{y} \right] + 2\alpha\mathbb{E}_Q \left[ \beta^T\mathbf{U}\mathbf{U}^T\mathbf{y} \right] + \mathbb{E}_Q \left[ \beta^T\beta \right]. \quad (16)$$

Compute Individual Expectations and using expectation properties:

$$\mathbb{E}_Q[\mathbf{y}^T\mathbf{y}] = \text{tr}(\mathbb{E}_Q[\mathbf{y}\mathbf{y}^T]). \quad (17)$$

Since:

$$\mathbb{E}_Q[\mathbf{y}\mathbf{y}^T] = \mathbf{U}\mathbf{U}^T + s^2 I + \mu_Q\mu_Q^T, \quad (18)$$

$$\mathbb{E}_Q[\mathbf{y}^T\mathbf{y}] = \text{tr}(\mathbf{U}\mathbf{U}^T) + s^2\text{tr}(I) + \text{tr}(\mu_Q\mu_Q^T), \quad (19)$$
$$= d + s^2 n + \|\mu_Q\|^2. \quad (20)$$

For the second expectation:

$$\mathbb{E}_Q[\mathbf{y}^T\mathbf{U}\mathbf{U}^T\mathbf{y}] = \text{tr}(\mathbf{U}\mathbf{U}^T\mathbb{E}_Q[\mathbf{y}\mathbf{y}^T]). \quad (21)$$

Substituting $\mathbb{E}_Q[\mathbf{y}\mathbf{y}^T]$:

$$\mathbb{E}_Q[\mathbf{y}^T\mathbf{U}\mathbf{U}^T\mathbf{y}] = \text{tr}(\mathbf{U}\mathbf{U}^T(\mathbf{U}\mathbf{U}^T + s^2 I + \mu_Q\mu_Q^T)), \quad (22)$$
$$= \text{tr}(\mathbf{U}\mathbf{U}^T) + s^2\text{tr}(\mathbf{U}\mathbf{U}^T) + \text{tr}(\mathbf{U}\mathbf{U}^T\mu_Q\mu_Q^T), \quad (23)$$
$$= d + s^2 d + \text{tr}(\mathbf{U}\mathbf{U}^T\mu_Q\mu_Q^T). \quad (24)$$

$$L_{SS}(\alpha, \beta) = (d + s^2 n + \|\mu_Q\|^2) - 2\alpha(d + s^2 d + \text{tr}(\mathbf{U}\mathbf{U}^T\mu_Q\mu_Q^T)) - 2\beta^T\mu_Q + \alpha^2 d + 2\alpha d\beta^T\mu_Q + \|\beta\|^2. \quad (25)$$

Combine each component, we can get

$$L_{SS}(\alpha, \beta) = d + s^2 n + \|\mu_Q\|^2 - 2\alpha(d + s^2 d + \|\mathbf{U}^T\mu_Q\|^2) + \alpha^2 d + 2\alpha d\beta^T\mu_Q - 2\beta^T\mu_Q + \|\beta\|^2 \quad (26)$$

Final Simplified Expression for this term

$$L_{SS}(\alpha, \beta) = s^2 n + (1 - \alpha)^2 d + (\alpha^2 - 2\alpha)s^2 d + \|\beta - \mu_Q\|^2 \quad (27)$$

**Expending second term:**

$$\frac{2\alpha d}{n-d} \mathbb{E}_Q \left[ \|(\mathbf{I} - \mathbf{U}\mathbf{U}^T)\mathbf{y}\|_2^2 \right]. \quad (28)$$

First, expanding the squared norm:

$$\|(\mathbf{I} - \mathbf{U}\mathbf{U}^T)\mathbf{y}\|_2^2 = \left((\mathbf{I} - \mathbf{U}\mathbf{U}^T)\mathbf{y}\right)^T \left((\mathbf{I} - \mathbf{U}\mathbf{U}^T)\mathbf{y}\right). \tag{29}$$

Since $(\mathbf{I} - \mathbf{U}\mathbf{U}^T)$ is symmetric:

$$= \mathbf{y}^T(\mathbf{I} - \mathbf{U}\mathbf{U}^T)\mathbf{y}. \tag{30}$$

Taking expectation:

$$\mathbb{E}_Q\left[\mathbf{y}^T(\mathbf{I} - \mathbf{U}\mathbf{U}^T)\mathbf{y}\right] = \operatorname{tr}\left((\mathbf{I} - \mathbf{U}\mathbf{U}^T)\mathbb{E}_Q[\mathbf{y}\mathbf{y}^T]\right). \tag{31}$$

Using the expectation property:

$$\mathbb{E}_Q[\mathbf{y}\mathbf{y}^T] = \mathbf{U}\mathbf{U}^T + s^2 I + \mu_Q\mu_Q^T. \tag{32}$$

Substituting:

$$= \operatorname{tr}\left((\mathbf{I} - \mathbf{U}\mathbf{U}^T)\left(\mathbf{U}\mathbf{U}^T + s^2 I + \mu_Q\mu_Q^T\right)\right). \tag{33}$$

Expanding the trace:

$$= \operatorname{tr}\left((\mathbf{I} - \mathbf{U}\mathbf{U}^T)s^2 I + (\mathbf{I} - \mathbf{U}\mathbf{U}^T)\mu_Q\mu_Q^T\right). \tag{34}$$

Since $(\mathbf{I} - \mathbf{U}\mathbf{U}^T)$ removes the $\mathbf{U}\mathbf{U}^T$ component:

$$= s^2(n - d) + \operatorname{tr}\left((\mathbf{I} - \mathbf{U}\mathbf{U}^T)\mu_Q\mu_Q^T\right). \tag{35}$$

Thus, the second term simplifies to:

$$\frac{2\alpha d}{n - d}\left[s^2(n - d) + \operatorname{tr}\left((\mathbf{I} - \mathbf{U}\mathbf{U}^T)\mu_Q\mu_Q^T\right)\right]. \tag{36}$$

Assuming $\mu_Q$ is entirely inside the subspace spanned by $\mathbf{U}$, the projection term vanishes, giving:

$$\frac{2\alpha d}{n - d}s^2(n - d). \tag{37}$$

Last, we take final simplification Now, simplifying the terms:

$$L_{SS}(\alpha, \beta) = s^2 n + (1 - \alpha)^2 d + (\alpha^2 - 2\alpha)s^2 d + \|\beta - \mu_Q\|^2 + 2\alpha d s^2. \tag{38}$$

Combining the $s^2 d$ terms:

$$(\alpha^2 - 2\alpha)s^2 d + 2\alpha s^2 d = \alpha^2 s^2 d - 2\alpha s^2 d + 2\alpha s^2 d = \alpha^2 s^2 d. \tag{39}$$

Thus, the final loss function simplifies to:

$$L_{SS}(\alpha, \beta) = s^2 n + (1 - \alpha)^2 d + \alpha^2 s^2 d + \|\beta - \mu_Q\|^2. \tag{40}$$

**Finally, we compute the derivatives**

For derivative with Respect to $\alpha$

$$\frac{\partial L_{SS}}{\partial \alpha} = -2d(1 - \alpha) + 2\alpha d s^2. \tag{41}$$

Setting this to zero and solving for $\alpha^*$:

$$\alpha^* = \frac{1}{1 + s^2}. \tag{42}$$

For derivative with respect to $\beta$

$$\frac{\partial L_{SS}}{\partial \beta} = 2(\beta - \mu_Q). \tag{43}$$

Setting this to zero and solving for $\beta^*$:

$$\beta^* = \mu_Q. \tag{44}$$

In conclusion,

1. $\alpha^*$ dynamically adjusts for noise variance shifts.
2. $\beta^*$ corrects for mean shifts, making adaptation robust in OOD settings.

## G   Supplementary Visualisations

While the main paper presents qualitative comparisons on UNet (FINE) under the anatomy shift, this appendix includes additional visualisations across the remaining domain shifts. We provide side-by-side comparisons of reconstruction results for our method and competing approaches, including DIP-TTT and ZS-SSL. Notably, ZS-SSL results are visualised under the VarNet backbone as originally proposed with data consistency block. Our visualisations offer a more comprehensive view of cross-domain performance across architectures and adaptation strategies.

### G.1   UNet

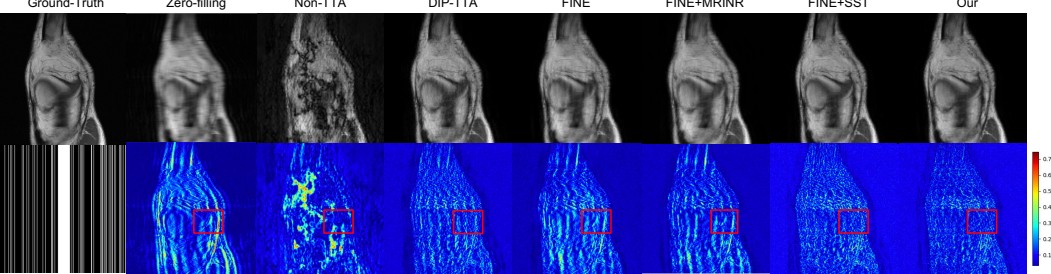

Figure 9: Comparison of different frameworks in UNet under dataset shift (Stanford to fastMRI) using the FINE method. The first row shows reconstructed MRI images, while the second row presents residual maps between reconstructions and full-sampled MRI.

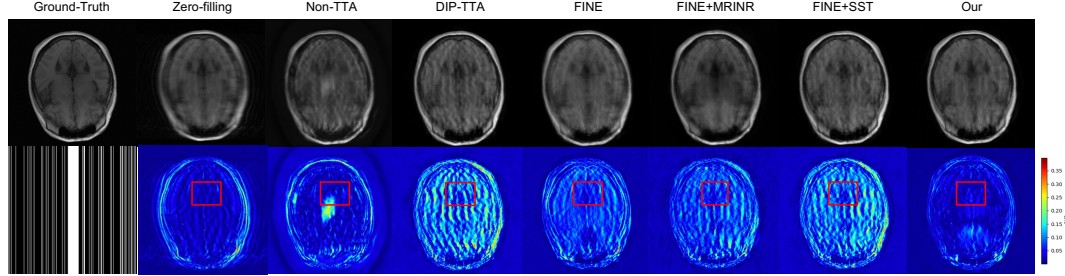

Figure 10: Comparison of different frameworks in UNet under modality shift (AXT2 to AXT1PRE) using the FINE method. The first row shows reconstructed MRI images, while the second row presents residual maps between reconstructions and full-sampled MRI.

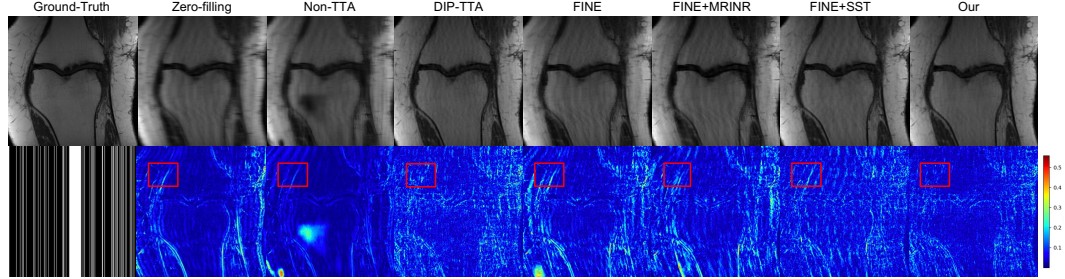

Figure 11: Comparison of different frameworks in UNet under acceleration shift (2X to 4X) using the FINE method. The first row shows reconstructed MRI images, while the second row presents residual maps between reconstructions and full-sampled MRI.

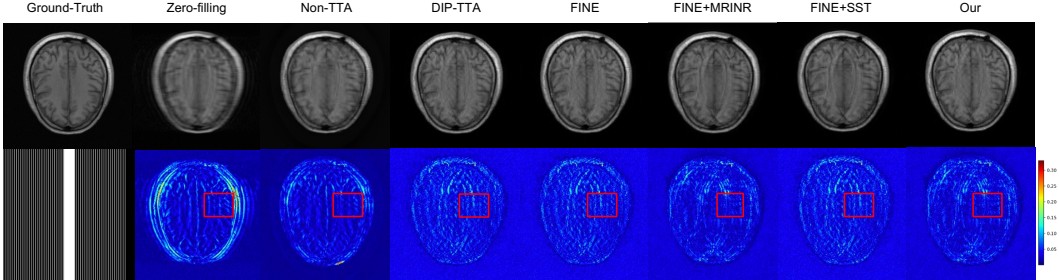

Figure 12: Comparison of different frameworks in UNet under sampling shift (random to uniform) using the FINE method. The first row shows reconstructed MRI images, while the second row presents residual maps between reconstructions and full-sampled MRI.

## G.2 VarNet

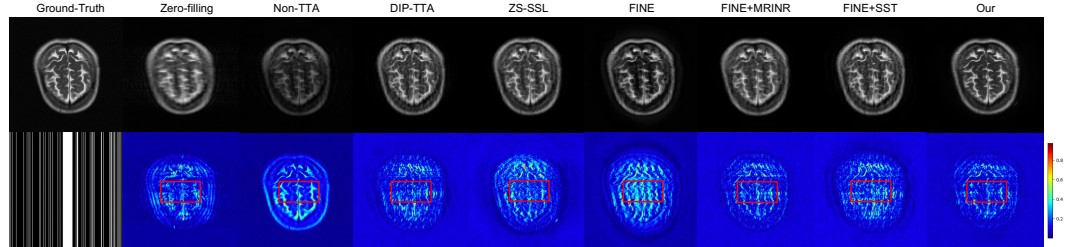

Figure 13: Comparison of different frameworks in VarNet under anatomy shift (Knee to Brain) using the FINE method. The first row shows reconstructed MRI images, while the second row presents residual maps between reconstructions and full-sampled MRI.

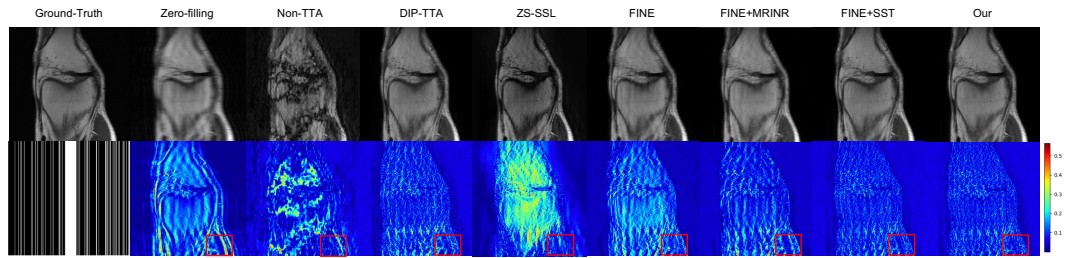

Figure 14: Comparison of different frameworks in VarNet under dataset shift (Stanford to fastMRI) using the FINE method. The first row shows reconstructed MRI images, while the second row presents residual maps between reconstructions and full-sampled MRI. The proposed method (far right) achieves the lowest residuals, indicating improved reconstruction accuracy

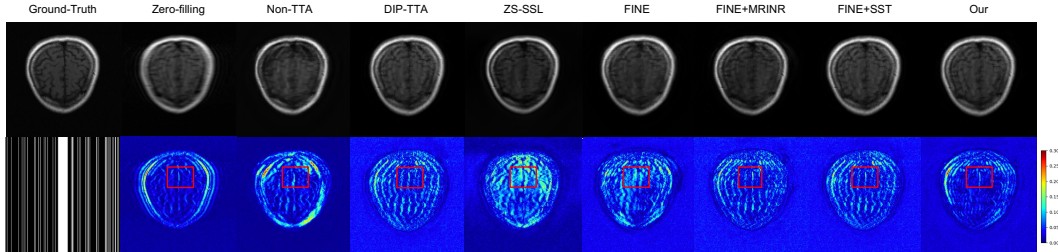

Figure 15: Comparison of different frameworks in VarNet under modality shift (AXT2 to AXT1PRE) using the FINE method. The first row shows reconstructed MRI images, while the second row presents residual maps between reconstructions and full-sampled MRI.

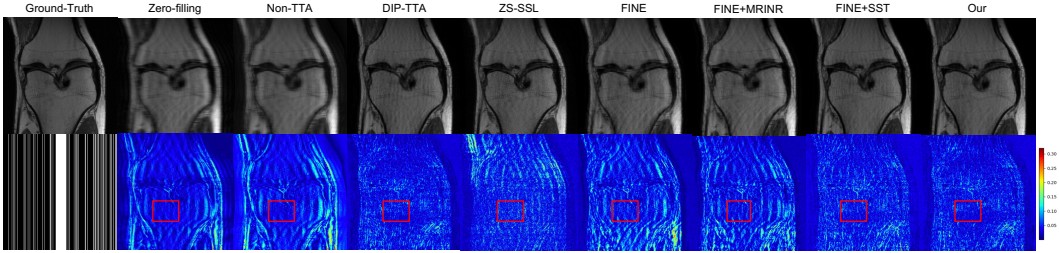

Figure 16: Comparison of different frameworks in Varnet under acceleration shift (2X to 4X) using the FINE method. The first row shows reconstructed MRI images, while the second row presents residual maps between reconstructions and full-sampled MRI.

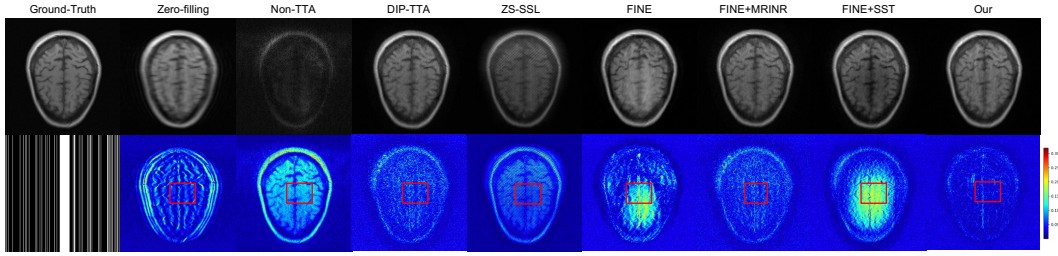

Figure 17: Comparison of different frameworks in VarNet under sampling shift (random to uniform) using the FINE method. The first row shows reconstructed MRI images, while the second row presents residual maps between reconstructions and full-sampled MRI.

