# OpenReview forum: "D2SA: Dual-Stage Distribution and Slice Adaptation for Efficient Test-Time Adaptation in MRI Reconstruction"
_NeurIPS.cc/2025/Conference — NeurIPS 2025 poster_

### Official Review · Reviewer_rEsT · 2025-06-27

**Clarity:** 2
**Significance:** 3
**Originality:** 2
**Rating:** 4
**Confidence:** 4

**Summary:**

This paper proposes D2SA, a two-stage test-time adaptation (TTA) framework for MRI reconstruction. The goal is to improve model performance when faced with distribution shifts between training and testing data. The method first employs a patient-wise adaptation stage using an MRI-specific Implicit Neural Representation (MR-INR) to capture global mean and variance shifts for a given patient. This is followed by a second stage that performs single-slice refinement. This refinement stage freezes most of the base network and the learned patient-level latent code for efficiency while fine-tuning the MR-INR and a novel, learnable Anisotropic Diffusion (AD) module to preserve structural details. The authors evaluate D2SA across five simulated domain shift scenarios, showing consistent performance improvements over several TTA baselines.

**Questions:**

1. The core of the paper is the two-stage design. Could you provide an ablation study that compares a simpler baseline (e.g., FINE) directly with that same baseline enhanced only by Stage 2 (i.e., FINE+SST+AD)? This would help clarify whether the complex Stage 1 patient-wise adaptation is truly necessary or if most of the benefit comes from the powerful slice-refinement stage alone.
2. The proposed system involves numerous components and corresponding hyperparameters (e.g., loss weights λ_self, λ_INR, λ_reg; latent code variance σ, etc.). The paper does not discuss the sensitivity of the results to these choices. How robust is D2SA to variations in these hyperparameters? A brief analysis would be crucial for assessing the method's practical usability and reproducibility.
Computational Overhead vs. Gain: While adaptation time is reported, what is the increase in parameter count and computational complexity (e.g., FLOPs) of the D2SA framework compared to a baseline like FINE? A more direct comparison would help readers assess if the marginal performance gains in some scenarios justify the added model and computational complexity

**Ethical Concerns:**

["NO or VERY MINOR ethics concerns only"]

**Final Justification:**

The authors have addressed most of my concerns, but it remains necessary to discuss their innovativeness from a methodological perspective. Therefore, I consider a score of 4 to be reasonable. If forced to choose between acceptance and rejection, I believe the paper can be published.

**Limitations:**

Yes

**Quality:**

3

**Strengths And Weaknesses:**

Strengths
1. The paper addresses the highly relevant and practical problem of domain shifts in medical imaging, which is a significant barrier to the real-world deployment of deep learning models.
2. The experimental setup is a clear strength. The authors have commendably evaluated their method across five different, well-motivated domain shift scenarios and against a suite of relevant TTA baselines.
3. The paper is well-structured and generally easy to read. The figures, particularly Figure 1 and Figure 2, effectively communicate the core concepts and the architecture of the proposed framework.

Weaknesses
1. The primary weakness is that the D2SA framework, while elaborate, appears to be a complex combination of several known techniques. Implicit Neural Representations (INRs), test-time feature modulation, self-supervised losses, and anisotropic diffusion are all existing concepts. While their combination is novel, the paper lacks a core conceptual breakthrough. The contribution feels more like a sophisticated engineering effort than a fundamental advance. The complexity of this multi-stage, multi-component system is very high, which may act as a barrier to adoption.
2. Due to the method's complexity, it is difficult to ascertain the primary drivers of the performance improvement. While the ablation studies are helpful, they don't fully disentangle the contributions. For example, the visual improvements shown in Figure 5 are attributed to the full "Our" method (FINE+MR-INR+SST+AD). It is unclear how much of this improvement is simply due to the powerful SST+AD refinement stage, versus the full two-stage pipeline. This makes it hard to judge the true value of the patient-wise adaptation stage.

---

> ### Author Rebuttal · Authors · 2025-07-30
>
> We sincerely thank the reviewer for the detailed summary and constructive comments. We appreciate your recognition of the practical value of our framework, the clarity of our experimental design, and the relevance of addressing domain shifts in medical imaging. Below, we respond to your main concerns regarding the framework, component contributions, hyperparameter sensitivity, and computational overhead, and we provide additional analyses to address the concerns.
>
> Q1:The Method’s Complexity and the Use of Known Components
>
> Conceptual contributions in adapting INR for test-time domain shift: To the best of our knowledge, INRs have not been previously used in test-time adaptation for image reconstruction. We leverage the compact functional expressiveness of INR to learn a new patient-specific representation at test time, not to generate images, but to guide the adaptation of source models in a structure-aware way. This is conceptually distinct from both conventional pretraining-based INR usage (e.g., Functa) and standard slice-wise TTA.
>
> Unified modular architecture with pluggable design: Each component (MR-INR, affine modulation, SST refinement, and AD module) is designed to be lightweight, interchangeable, and orthogonal. This enhances not only performance, but also flexibility and practical adoption. For example:
>
> If compute is limited, users can skip Stage 2 and rely on Stage 1 alone.
>
> If specific slices are flagged by clinicians, only Stage 2 is applied.
>
> A holistic test-time design perspective: Our framework proposes a structured pipeline that mimics how real clinical workflows operate—first broadly adapting to the patient, then selectively refining critical regions. This hierarchical control is not present in prior works and reflects a pragmatic yet theoretically motivated design.
>
> While we do build on existing components, our architectural integration and adaptation strategy are novel, practical, and aligned with real-world test-time usage constraints—thus providing both scientific insight and engineering utility.
>
> Q2:Necessity of Stage 1: Is most benefit from Stage 2?
>
> We provide a detailed ablation comparing three configurations on both UNet and VarNet under anatomy shift, including Stage 2 only (FINE + SST + AD); Stage 1 + Stage 2 without MR-INR (FINE + SST + AD but with pretrained encoder from FINE) and Full D2SA (FINE + MR-INR + SST + AD).
>
> | Configuration (UNet)                           | PSNR ↑    | Params (M) ↓ | Time (min/patient) ↓ |
> | ---------------------------------------- | --------- | ------------ | -------------------- |
> | Stage 2 only (FINE + SST + AD)           | 27.31     | 17.62        | 18.7                 |
> | Stage 1 + 2 w/o MR-INR (FINE + SST + AD) | 27.37     | 17.62        | 22.6                 |
> | **Full D2SA**                            | **27.71** | 17.89        | **12.1**             |
>
> | Configuration  (VarNet)                          | PSNR ↑    | Params (M) ↓ | Time (min/patient) ↓ |
> | ---------------------------------------- | --------- | ------------ | -------------------- |
> | Stage 2 only (FINE + SST + AD)           | 27.50     | 16.74        | 56.8                 |
> | Stage 1 + 2 w/o MR-INR (FINE + SST + AD) | 27.58     | 16.74        | 53.3                 |
> | **Full D2SA**                            | **27.68** | 16.98        | **17.1**             |
>
> Compared to Stage 2 only (FINE+SST+AD), which uses only FINE features, our Full D2SA introduces the Stage 1 (MR-INR) and consistently boosts PSNR by 0.18~0.4 dB More importantly, inference time is reduced by >30% (UNet) and nearly 3× (VarNet) when MR-INR provides better initialization for SST. This shows that Stage 1 is not redundant: it not only improves image quality but also accelerates slice-level optimization in Stage 2, especially combining MR-INR in the stage 1.
>
> Q3: Hyperparameter sensitivity
>
> To further support the robustness of our method in hyperparameters settings, we show the dedicated robustness results on the key hyperparameters in both Stage 1 (MR-INR) and Stage 2 (SST+AD), including: Latent code size (64 / 128 / 256)；INR network depth (3 / 4 / 5)；Latent code initialization std (0.1 / 0.01 / 0.001)
>
> | TTA Stage   | Variant                           | SSIM↑                  | PSNR↑                  | LPIPS↓                 |
> | ----------- | --------------------------------- | --------------------- | --------------------- | --------------------- |
> | **Stage 1** | Latent Code Size (64 / **128** / 256) | 0.849 / 0.845 / 0.850 | 26.39 / 26.37 / 26.47 | 0.345 / 0.346 / 0.335 |
> |             | INR Depth (3 / **4** / 5)             | 0.842 / 0.845 / 0.848 | 26.34 / 26.37 / 26.44 | 0.345 / 0.346 / 0.346 |
> |             | Init Std (0.1 / **0.01** / 0.001)     | 0.845 / 0.845 / 0.847 | 26.50 / 26.37 / 26.37 | 0.346 / 0.346 / 0.345 |
> | **Stage 2** | Latent Code Size (64 / **128** / 256) | 0.876 / 0.876 / 0.877 | 27.53 / 27.71 / 27.53 | 0.323 / 0.320 / 0.320 |
> |             | INR Depth (3 / **4** / 5)             | 0.874 / 0.876 / 0.877 | 27.54 / 27.71 / 27.56 | 0.321 / 0.320 / 0.319 |
> |             | Init Std (0.1 / **0.01** / 0.001)     | 0.876 / 0.876 / 0.875 | 27.53 / 27.71 / 27.52 | 0.321 / 0.320 / 0.322 |
>
> As shown above, the performance across PSNR / SSIM / LPIPS remains stable, with <0.15 dB variation in PSNR and negligible impact on LPIPS.
>
> We also show the results about different combinations of loss weights for Stage 1: λ_Self / λ_INR / λ_Reg; Stage 2: λ_Self / λ_INR.
>
> | Stage   | λ Settings     | PSNR ↑ | SSIM ↑ | LPIPS ↓ |
> | ------- | -------------- | ------ | ------ | ------- |
> | Stage 1 | 1 / 1 / 1e-4   | 24.97  | 0.832  | 0.385   |
> |         | 1 / 0.1 / 1e-4 | 23.85  | 0.823  | 0.386   |
> |         | 1 / 1 / 0.1    | 24.00  | 0.822  | 0.388   |
> | Stage 2 | 1 / 1          | 28.04  | 0.860  | 0.287   |
> |         | 1 / 0.1        | 28.00  | 0.861  | 0.356   |
> |         | 1 / 0.01       | 27.99  | 0.861  | 0.350   |
>
> We used empirically chosen default settings (Stage 1: λ = (1, 1, 1e-4); Stage 2: λ = (1, 1)). The results remain stable across a range of values, indicating good robustness of our framework. However, we observe that setting λ_INR too low (e.g., 0.1) leads to performance degradation, particularly in Stage 1 and LPIPS in stage 2. This is expected, as MR-INR plays a central role in modeling patient-level distribution shifts. A sufficiently weighted INR loss encourages better representation learning of domain-specific patterns, which is crucial for boosting TTA effectiveness.
>
>
> Q4:Computational Overhead vs. Gain:
>
> While our method does introduce additional modules (INR and AD), we would like to clarify that we design the D2SA with practical efficiency and controllability in mind:
>
> Frozen base networks reduce trainable parameters: Unlike methods like FINE that update all weights, our approach freezes the base CNN (U-Net/VarNet), and only updates light-weight modules (MR-INR, affine modulation, AD), leading to significantly fewer trainable parameters. This is evident in Table 2, where D2SA consistently requires fewer parameters than full-model tuning.
>
> Stage-wise adaptation reduces convergence time: Although D2SA has more components, it achieves faster and more stable convergence. As shown in Figure 4 (main) and Figure 7 (supplement), inference time is reduced at least 60% compared to FINE+SST or DIP-TTT under the same early stopping setup.
>
> Efficient initialization for refinement: Importantly, Stage 1 acts as a structured, patient-aware initialization for Stage 2, providing prior distribution alignment and faster convergence for slice-level refinement. This makes Stage 2 more effective and efficient, especially for structurally important slices.
>
> In summary, D2SA offers a favorable trade-off: while FLOPs do increase, our design leads to lower parameter counts, faster convergence, and more robust reconstructions, making it more practical for clinical deployment than existing monolithic TTA approaches.

---

> > ### Comment · Reviewer_rEsT · 2025-08-06
> >
> > I would like to thank the authors for their detailed rebuttal, which effectively addressed my concerns. I have raised my score.

---

> > > ### Author Response · Authors · 2025-08-06
> > > **Appreciation for Updated Assessment**
> > >
> > > Thank you for taking the time to read our rebuttal and for your constructive feedback throughout the process. We truly appreciate your updated assessment and will incorporate all your suggestions in the final paper and appendix.
> > >
> > > .

---

> ### Author Response · Authors · 2025-08-05
> **Further Discussion Welcome**
>
> Thank you again for your thoughtful review. We would greatly appreciate it if you could share any further thoughts regarding our rebuttal. Your feedback would be invaluable in helping us improve the final version.

---

### Official Review · Reviewer_SZTm · 2025-07-02

**Clarity:** 3
**Significance:** 3
**Originality:** 3
**Rating:** 5
**Confidence:** 3

**Summary:**

The paper proposes a dual-stage Test-Time Adaptation (TTA) method (D2SA) to address domain shifts in MRI reconstruction.

Stage 1 (distribution-level adaptation) uses MRI implicit neural representations (MR-INR) to adapt to patient-specific distribution shifts.

Stage 2 (slice-level adaptation) employs anisotropic diffusion modules to refine reconstructions at the slice level, preserving structural fidelity.

It demonstrates consistent improvements across various MRI distribution shifts (anatomy, modality, dataset, sampling, and acceleration).

**Questions:**

Can the authors clarify explicitly what distinguishes their method fundamentally from existing slice-wise refinement or TTA methods, beyond merely combining MR-INR and anisotropic diffusion?

How sensitive is the method to hyperparameter choices such as latent code dimensionality or anisotropic diffusion parameters?

**Ethical Concerns:**

["NO or VERY MINOR ethics concerns only"]

**Final Justification:**

Technique-wise, this paper sounds reasonable to me.

**Limitations:**

yes

**Quality:**

3

**Strengths And Weaknesses:**

Strengths:

**Good experimental validation**

The paper presents extensive quantitative evaluations under multiple distribution shifts, demonstrating performance improvements. Visual results provided in the appendix are convincing and clearly show the effectiveness of the proposed method in comparison to existing baselines like DIP-TTT and ZS-SSL.

**Effective combination of existing techniques**
Integrates Implicit Neural Representations (INR) and anisotropic diffusion, both proven techniques, to enhance reconstruction performance and efficiency. It achieves notable efficiency improvements (reduced adaptation time compared to ZS-SSL and DIP-TTT), making the method practical for real-world scenarios.


Weakness

x. The authors acknowledge inspirations from existing works (e.g., DIP-TTT, Functa, and SSDU), but the paper could explicitly discuss the novelty more clearly, especially distinguishing from recent diffusion-based approaches.

x. The authors introduce an additional layer of complexity by using the INR framework to handle patient-level distribution adaptation.
However, INR requires careful hyperparameter tuning (latent vector size, network depth, initialization) to avoid instability issues. Hence, a clearer analysis of hyperparameter sensitivity and computational complexity could improve transparency and could help understand the practical implications of this complexity better.

---

> ### Author Rebuttal · Authors · 2025-07-30
>
> We thank the reviewer for the thoughtful suggestions. Your recognition of our dual-stage framework’s effectiveness and efficiency is greatly appreciated. Below, we address your comments regarding the clarity of architectural differences, the distinctions from diffusion-based test-time adaptation methods, and hyperparameter sensitivity.
>
> Q1: Architectural Differences
>
> Beyond architectural differences, our two-stage D2SA framework is designed with realistic clinical deployment in mind, where direct zero-shot transfer or pure slice-wise refinement often fails to provide reliable reconstructions—especially under large distribution shifts. This unreliability can directly impact early-stage screening and the localization of small or subtle lesions.
>
> In clinical scenarios, radiologists often need initial high-quality reconstructions across all slices for efficient review, followed by selective refinement on suspicious or diagnostically relevant slices.
>
> To this end:
>
> Stage 1 (MR-INR) performs a fast, patient-level adaptation, leveraging inter-slice structure to learn global distributional shifts (mean, variance). It provides a reliable reconstruction baseline within ~5–7 minutes, without requiring slice-wise optimization.
>
> Stage 2 (SST + AD module) then allows targeted refinement on demand, enhancing structural fidelity on specific slices without re-training the full network. The AD module preserves fine-grained anatomical details (e.g., tissue boundaries), which are often over-smoothed by conventional TTA methods.
>
> Each component is deliberately positioned:
>
> MR-INR enables compact patient-specific modeling, aligned with the notion of shared anatomical priors.
>
> Affine adaptation (α, β) efficiently compensates for patient-specific mean/variance shifts with provable benefits (Section 4.3).
>
> The AD module, integrated into SST, provides structure-aware filtering under frozen CNNs, striking a balance between efficiency and precision.
>
> Q2: The distinctions from diffusion-based test-time adaptation methods
>
> Unlike diffusion-based TTA approaches that require heavy pretraining and offer limited flexibility, our method is pluggable, interpretable, and clinically actionable. This hierarchical strategy is not merely a module combination but a functionally structured design tailored for practical, slice-selective, low-resource test-time adaptation in real-world medical workflows.
>
> Q3: Sensitivity about hyperparameter on latent code and AD
>
> To clarify robustness of our INR-based adaptation, we show additional sensitivity experiments on three critical hyperparameters: latent code size, INR network depth, and latent code initialization scale. As shown below, our method exhibits stable performance across a wide range of values:
>
> | TTA Stage   | Variant                           | SSIM↑                  | PSNR↑                  | LPIPS↓                 |
> | ----------- | --------------------------------- | --------------------- | --------------------- | --------------------- |
> | **Stage 1** | Latent Code Size (64 / **128** / 256) | 0.849 / 0.845 / 0.850 | 26.39 / 26.37 / 26.47 | 0.345 / 0.346 / 0.335 |
> |             | INR Depth (3 / **4** / 5)             | 0.842 / 0.845 / 0.848 | 26.34 / 26.37 / 26.44 | 0.345 / 0.346 / 0.346 |
> |             | Init Std (0.1 / **0.01** / 0.001)     | 0.845 / 0.845 / 0.847 | 26.50 / 26.37 / 26.37 | 0.346 / 0.346 / 0.345 |
> | **Stage 2** | Latent Code Size (64 / **128** / 256) | 0.876 / 0.876 / 0.877 | 27.53 / 27.71 / 27.53 | 0.323 / 0.320 / 0.320 |
> |             | INR Depth (3 / **4** / 5)             | 0.874 / 0.876 / 0.877 | 27.54 / 27.71 / 27.56 | 0.321 / 0.320 / 0.319 |
> |             | Init Std (0.1 / **0.01** / 0.001)     | 0.876 / 0.876 / 0.875 | 27.53 / 27.71 / 27.52 | 0.321 / 0.320 / 0.322 |
>
> Our default configuration (highlighted in bold: 128-d latent code, depth 4, init std 0.01) achieves a strong trade-off between stability and performance. Performance is consistent across hyperparameter ranges, with differences ≤ 0.02 PSNR or ≤ 0.01 LPIPS. This suggests our method is not overly sensitive and avoids instability even with pluggable configurations. The default setup ensures robust adaptation without excessive tuning overhead.
>
> Moreover, to show the robustness of our method to the anisotropic diffusion (AD) step size, we conducted ablation experiments under modality shift (UNet with FINE backbone). As shown below, varying the AD step size from 1.0 to 0.01 leads to minimal changes in performance:
>
> | AD Step Size | PSNR ↑    | SSIM ↑    | LPIPS ↓   |
> | ------------ | --------- | --------- | --------- |
> | **1.0**      | **27.71** | **0.876** | **0.320** |
> | 0.1          | 27.38     | 0.874     | 0.322     |
> | 0.01         | 27.36     | 0.874     | 0.323     |
>
> These results demonstrate that our method is robust to the choice of the diffusion step size, with less than 0.4 dB PSNR fluctuation and negligible variation in perceptual quality. The default choice of step=1.0 achieves the best trade-off between sharpness and stability.

---

> ### Author Response · Authors · 2025-08-05
> **Follow-up on Rebuttal**
>
> We truly appreciate your initial feedback. If you have any further comments on our rebuttal, we’d be grateful to hear them during the discussion phase. Thank you again for your time and consideration.

---

> > ### Comment · Reviewer_SZTm · 2025-08-05
> >
> > The new ablation study result looks great! Thank you!

---

> > > ### Author Response · Authors · 2025-08-06
> > > **Appreciation for Your Feedback**
> > >
> > > Thank you for your positive feedback and earlier suggestions, which directly motivated the improvements we made. Your comments and insights will be incorporated into the final manuscript and supplementary material.

---

### Official Review · Reviewer_5sGP · 2025-07-02

**Clarity:** 2
**Significance:** 3
**Originality:** 3
**Rating:** 4
**Confidence:** 3

**Summary:**

The authors propose a dual-stage test-time adaptation framework for fMRI reconstruction in the compressed sensing domain. The first stage is to adapt a model trained on one domain to a new patient more generally, and the second refines the model to the individual slices of that patient. Ablation studies and results across five datasets indicate that the each of the authors' stages and modules contributes to performance improvements in terms of SSIM and PSNR.

**Questions:**

As mentioned in the strengths and weaknesses section, I think the following questions are important to answer:
1. Why are the ablations purely done on the U-Net architecture when both architectures were used in the paper?
2. Why did the authors choose SIREN when the paper they cite seems to imply that other architectures could have been better choices?
3. Are the results significant?

**Ethical Concerns:**

["NO or VERY MINOR ethics concerns only"]

**Final Justification:**

The authors have generally been able to address my concerns in their rebuttal, it is hard to verify that they have improved their figures, and exactly how much better they explain their method, but I believe (given the examples of improved writing they give in their rebuttal) that the authors will be able to make the changes that I find important before recommending this paper for publication. Given that I can't fully assess the above, I decided to increase my score to a 4.

**Limitations:**

Yes

**Quality:**

1

**Strengths And Weaknesses:**

**Strengths**\
I think the authors introduce original modifications to the field of test-time adaptation, I like the idea of separating out the patient adaptation and single-slice refinement. Moreover, the authors' results seem to indicate a significant improvement over previous works, although I would think it strengthens the work even more if the authors actually performed significance tests for their results. Especially compared to DIP-TTT, some of the results come close, and it is unclear whether these results are significant. All-in-all, I think this work can be a significant contribution to the field if the authors address the issues below in their rebuttal.

**Major weaknesses**\
I think the major weakness for this paper is that the quality and clarity need to be significantly improved before I can recommend this paper for acceptance. The introduction of the method is quite unclear, and I found it hard to follow exactly how all of the authors' improvements fit together into a larger framework. Specifically, the explanation of the AD module and section 4.2&4.3 are unclear. An example of an unclear paragraph are lines 124-128. Moreover, this extends to Figure 4, which is incredibly hard to read, is badly formatted, and I can therefore not verify their U-Net results. It is also unclear to me why the results are presented differently for the VarNet (table 2) and U-net architecture (Figure 4). Moreover, why are the ablations purely done on the U-Net architecture when both architectures were used in the paper?

**Minor weaknesses**\
The authors cite the following paper: "Among various designs [15], SIREN remains a strong choice ...". However, in the paper that the authors refer to, the discussion states the following:
"For applications where computational efficiency is less critical and the highest reconstruction quality is
needed, Incode would be the preferred choice. However, for real-time applications or cases where a balance
between quality and speed is crucial, Fr and Finer offer strong alternatives with their adaptable frequency
handling and more manageable computation times. Siren can be a good option when tasks require periodic
signal reconstruction but with less variability in frequency content. One area where INR methods still face
challenges is scalability, particularly in handling extremely high-resolution or highly detailed tasks. Future
work could explore more efficient frequency encoding mechanisms that reduce the computational overhead
without sacrificing quality. Additionally, optimising activation functions to be more adaptive to task-specific
requirements could improve generalisation across diverse datasets and applications. Enhancing the dynamic
adaptability of models like Finer could allow for more robust performance across tasks that involve varying
levels of detail, potentially improving their usability in real-world scenarios."
It seems like INCODE, Fr, and Finer are potentially better backbone models for this specific task because I am unsure why we would expect more periodic signal reconstruction, but less variability in the frequency content. Moreover, in their paper, (Table 2), the medical imaging results indicate that Fr would be a much better choice. Can the authors comment on this or potentially perform an ablation study to test these effects?

**Spelling/grammar**\
L3: "... this discrepancies ..." -> these \
L15-16: "... can integrate well with various self-supervised (SSL) framework, ..." -> frameworks \
L126: "... and Fourier feature ..." -> features \
L138: "... before last layer ..." -> the last layer \
L166: "... given an set of feature u, ..." -> given a set of features u \
L181: "In calculation of AD equation ..." -> the AD equation

---

> ### Author Rebuttal · Authors · 2025-07-30
>
> We sincerely thank the reviewer for the detailed and constructive feedback. Your comments have greatly helped us identify key areas for the clarification of our method and results. For the spelling/grammar error, we will correct them in final version. Below, we address your main concerns.
>
> Q1: Clarity Concerns
>
> In the Introduction (Lines 47–70) and the beginning of Section 3 (Method), we summarize the motivation and high-level design of D2SA, highlighting its two-stage structure:
>
> Stage 1: Patient-wise Distribution Adaptation – We use an MR-specific Implicit Neural Representation (MR-INR) to model shared representations across slices. Each slice’s coordinates are extended with a patient-specific latent code (z_p) and random Fourier features to stabilize convergence and express frequency shifts. These inputs are passed through a SIREN network to learn mean/variance modulation parameters (α,β), allowing distribution alignment for all slices in a patient.
>
> Stage 2: Slice-wise Structural Refinement – For each slice, we freeze the encoder and patient latent code and fine-tune only the INR decoder and a lightweight, learnable Anisotropic Diffusion (AD) module. The AD module extracts directional gradients using difference convolutions (see Line 171–180) and generates adaptive edge-aware refinement weights.
>
> Details of the AD module in Section 4.2&4.3: Regarding Section 4.2, due to space constraints, we initially omitted the explicit formulation of the five directional difference convolutions used to estimate gradient-aware anisotropic diffusion coefficients. These convolutions extract directional derivatives (horizontal, vertical, diagonal, etc.) with end-to-end learning machanism. For example, taking the Horizontal Difference Convolution (HDC) as a case: the horizontal gradient is first computed via differences between pixel pairs along the x-axis. After training, the learned kernel weights are re-arranged and applied directly as convolution filters on the input feature map. These filters exhibit properties similar to handcrafted operators (e.g., their horizontal weights sum to zero), enabling structural-aware modulation. We will briefly summarize this process in the main paper. For Section 4.3, the theoretical derivation of the LSS-based optimization is given in Appendix Section 5; we will highlight this in the main text.
>
> Clarifying Lines 124–128: We rephrase it as follows for better clarity: “Following patient-level batch training, each slice is associated with a latent identity code. In the MR-INR branch, this latent code is concatenated with Fourier-encoded spatial coordinates before being passed to a SIREN network. This combination enhances convergence and representation power by encoding slice-specific anatomical priors and capturing periodic signals in MRI signal acquisition.”
>
> Figure 4 and UNet result：To avoid table overcrowding and emphasize key trends, we visualized UNet results as charts. These highlight that: (i) MR-INR provides the largest PSNR gain under modality shift (top-left); (ii) our full method outperforms baselines with lower LPIPS and faster inference (bottom-left); (iii) MR-INR and SST consistently improve SSIM across all domain shifts (right). Full UNet tables are in Appendix Table 1, and VarNet charts in Appendix Fig. 1. We will move UNet table to main page and keep all charts in Appendix in the final version.
>
>
> Q2: Ablation studies on VarNet
>
> We extend the full-stage ablations to VarNet under the anatomy shift setting, including additional baselines including SST-only+AD and FINE+SST+AD. The VarNet part will be added into final version.
>
> | MR-INR / SST Ablation (VarNet)                                        | PSNR ↑ | Params ↓ | Time ↓ |
> |------------------------------------------------|--------|-----------|--------|
> | Patient-wise training                          |        |           |        |
> | FINE                                           | 24.01  | 29.45     | 3.9    |
> | +MR-INR (🧊 latent)                             | 26.37  | 29.69     | 4.7    |
> | +MR-INR (🔥 latent)                             | 26.45  | 29.69     | 5.0    |
> | Single-slice training without stage 1          |        |           |        |
> | SST (🧊 CNN + 🔥 AD)                             | 27.50  | 16.74     | 56.8   |
> | Single-slice training with stage 1             |        |           |        |
> | FINE + SST (🔥 CNN)                             | 27.57  | 29.45     | 53.6   |
> | FINE + SST (🧊 CNN + 🔥 AD)                      | 27.58  | 16.74     | 53.3   |
> | MR-INR + SST (🔥 CNN)                           | 27.68  | 29.69     | 20.9   |
> | MR-INR + SST (🔥 CNN + 🔥 AD)                   | 27.61  | 43.79     | 34.0   |
> | MR-INR🧊 + SST (🧊 CNN + 🔥 AD)                  | 27.63  | 16.74     | 27.8   |
> | MR-INR + SST (🧊 CNN)                           | 27.45  | **2.88**  | 21.6   |
> | MR-INR + SST (🧊 CNN + 🔥 AD) (**Ours**)        | **27.68** | 16.98 | **17.1** |
>
> The extended ablation study highlights the functional contributions and trade-offs of each D2SA component across VarNet backbones. We observe that Stage 2 (SST+AD) with/without stage1 training offer reasonable performance, but at the cost of significantly higher inference time (~50–56 mins per patient). Incorporating Stage 1 (MR-INR) leads to stronger initialization for slice-wise refinement, reducing convergence time substantially (up to 3× faster inference) and yielding consistent performance gains. Notably, our final configuration (MR-INR (🧊 latent) + SST (🧊 CNN + 🔥 AD) achieves the best PSNR while maintaining low parameter count and the fastest inference time (e.g., 17.1 min/patient for VarNet). These results validate our core claim: D2SA’s staged modularity is not only conceptually sound, but also practically effective, enabling scalable, fast, and adaptive inference with minimal tuning. Each module remains plug-and-play, ensuring generalizability and ease of adoption in various architectures and deployment scenarios.
>
> Q3:Choice of SIREN over Other INRs
>
> To clarify our choice of SIREN as the INR backbone in MR-INR, we conducted an additional ablation comparing different INR backbones under both patient-wise (FINE+MR-INR) and single-slice (FINE+MR-INR+SST) training. Results are summarized below:
>
> | Stage 1 | Stage 1 (SSIM↑ / PSNR↑ / LPIPS↓) | Stage1 + Stage 2 | Stage 2 (SSIM↑ / PSNR↑ / LPIPS↓) |
> | :--- | :--- | :--- | :--- |
> | WIRE | 0.839 / 25.78 / 0.354 | Finer | 0.873 / 27.40 / 0.323 |
> | Finer | 0.845 / 26.05 / 0.350 | WIRE | 0.872 / 27.44 / 0.321 |
> | Random PE + MLP | 0.845 / 25.76 / 0.349 | Random PE + MLP | 0.864 / 27.22 / 0.326 |
> | Only SIREN | 0.845 / 26.16 / 0.349 | Only SIREN | 0.874 / 27.51 / 0.322 |
> | **Ours (PE + SIREN)** | **0.845 / 26.37 / 0.346** | **Ours (PE + SIREN)** | **0.876 / 27.71 / 0.320** |
>
> Our design combining random Fourier positional encoding with SIREN consistently yields the best or near-best performance, especially in perceptual quality (LPIPS). While Finer offers frequency adaptability, its more complex architecture increases overhead without clear performance gains; WIRE, though efficient, suffers from unstable convergence and limited representation in medical shift settings. Compared to INCODE, our SIREN-based MR-INR avoids the need for pretraining, supporting our lightweight, plug-and-play adaptation objective.
>
> We will include these results in the final paper.
>
> Q4:Result Significance and Efficiency
>
> We thank the reviewer for the helpful suggestion. We would like to clarify statistical significance tests are already included in Supplementary Figure 8, where we compare our method (SSDU+MR-INR+SST) with DIP-TTT and +SST using Wilcoxon signed-rank tests. Under UNet (acceleration shift) and VarNet (sampling shift), we observe significant improvements in all metrics (p value < 0.05).
>
> We also would like to highlight efficiency gains, even in cases with close average scores. For example:
>
> In VarNet on anatomy shift (Table 1), our method improves SSIM from 0.878 → 0.882 and PSNR from 27.67 → 27.68, while reducing inference time from 52.5 min to 17.1 min, a 67% reduction.
>
> In U-Net on sampling shift (Table 3), our method achieves LPIPS 0.311 vs. 0.314 (DIP-TTT) with 13.2 min vs. 27.1 min, yielding a>50% faster adaptation while maintaining superior quality.

---

> > ### Comment · Reviewer_5sGP · 2025-08-03
> > **Response to the authors**
> >
> > Thank you for your thorough rebuttal to my original concerns. It is important to me that the authors commit to improving the figures (even if they are moved to the Appendix), and the readability of the manuscript for the camera-ready version. Moreover, it would be beneficial if the authors can include significance results in Table 1. Other than that, I want to thank the authors for addressing my concerns, and I have increased my score to reflect their improvements of the manuscript.

---

> > > ### Author Response · Authors · 2025-08-04
> > > **Appreciate the thoughtful feedback**
> > >
> > > Thank you very much for your thoughtful feedback that really helped to improve this paper and we also thank for increasing your score. All suggestions will be reflected in the final version.

---

### Note · Authors · 2025-08-13

We sincerely thank all reviewers and the Area Chair for their time, thoughtful feedback, and engagement throughout the review process. All reviewers acknowledged that their concerns were thoroughly addressed, and all explicitly noted improvements and raised their scores accordingly. We deeply appreciate this positive and constructive dialogue.

We are committed to incorporating all promised improvements into the final version of the paper to ensure clarity and completeness. Thank you again for your thoughtful evaluation and support. We hope these positive outcomes will contribute to a favorable recommendation. We also extend our sincere appreciation to the Area Chair for the time and effort dedicated to managing our submission.

---

### Decision · Program_Chairs · 2025-09-17

**Decision:**

Accept (poster)

**Comment:**

This paper proposes D2SA, a dual-stage test-time adaptation framework for MRI reconstruction, designed to improve robustness under distribution shifts. The method is evaluated across five scenarios: anatomy, modality, acceleration, SNR, and sampling, showing consistent gains over existing TTA baselines.
Reviewers highlighted key strengths, including the clinical importance of tackling domain shifts, the comprehensive experimental evaluation, and the clear presentation. At the same time, they noted that the contribution feels incremental as it combines existing components. There was also a concern about the complexity of the framework, and it was initially unclear which elements drive the improvements or how robust the method is to hyperparameters and overhead.
The rebuttal provided clarifications, with additional ablations, robustness analyses, and efficiency results that helped address concerns about necessity, stability, and practicality.
Overall, while the conceptual novelty remains limited, the work is technically solid, carefully validated, and practically relevant. The reviewers and myself  consider this a reasonable accept.